# Revisiting Prompt Optimization with Large Reasoning Models—A Case Study on Event Extraction

## Abstract

Large Reasoning Models (LRMs) such as DeepSeek-R1 and OpenAI o1 have demonstrated remarkable capabilities in various reasoning tasks. Their strong capability to generate and reason over intermediate thoughts has also led to arguments that they may no longer require extensive prompt engineering or optimization to interpret human instructions and produce accurate outputs. In this work, we aim to systematically study this open question, using the structured task of event extraction for a case study. We experimented with two LRMs (DeepSeek-R1 and o1) and two general-purpose Large Language Models (LLMs) (GPT-4o and GPT-4.5), when they were used as task models or prompt optimizers. Our results show that on tasks as complicated as event extraction, LRMs as task models still benefit from prompt optimization, and that using LRMs as prompt optimizers yields more effective prompts. Our finding also generalizes to tasks beyond event extraction. Finally, we provide an error analysis of common errors made by LRMs and highlight the stability and consistency of LRMs in refining task instructions and event guidelines.

## 1 Introduction

In recent years, Large Language Models (LLMs) have demonstrated remarkable capabilities across various natural language processing tasks. However, their proficiency in complex reasoning tasks has often been limited (Zhou et al., 2022). To address this, a new class of models, known as Large Reasoning Models (LRMs), has emerged, focusing on enhancing reasoning abilities through advanced training methodologies. Two prominent examples are DeepSeek-R1 (Guo et al., 2025) and OpenAI's o1 (Zhong et al., 2024), both setting new standards in various reasoning tasks.

The advent of these advanced reasoning models has sparked discussions (Wang et al., 2024a; OpenAI, 2025; Mantaras, 2025; Together AI, 2025; Menendez et al., 2025) about the necessity of prompt optimization—the process of refining

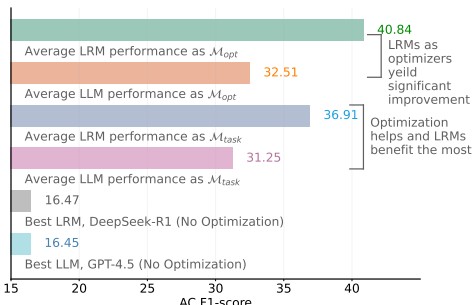

Figure 1: Summary of our main results, where LRMs and LLMs are used as either the task model ($\mathcal{M}_{task}$) or the optimizer ($\mathcal{M}_{opt}$) in prompt optimization, and we observed a strong advantage of LRMs over LLMs.

input prompts to guide model outputs effectively (Zhou et al., 2022; Yang et al., 2024; Srivastava et al., 2024; Agarwal et al., 2024; Guo et al., 2024; Fernando et al., 2024; Li et al., 2025). Traditionally, prompt optimization has been crucial for enhancing LLM performance, with frameworks like PromptAgent (Wang et al., 2024b) and OPRO (Yang et al., 2024) automating the creation and refinement of prompts through iterative feedback and strategic planning. However, the inherent reasoning capabilities of LRMs raise questions about whether such prompt optimization techniques are equally beneficial for these models. While previous studies have demonstrated the effectiveness of prompt optimization in improving LLM performance, there is a notable gap in research focusing on its impact on LRMs. Moreover, many existing prompt optimization studies focus on tasks where zero-shot

baselines already perform well, whereas recent work, such as Gao et al. (2024), demonstrates that even powerful models like GPT-4 struggle with Information Extraction tasks, underscoring the need for more targeted and optimized prompting strategies.

To fill this gap, we conduct the first systematic study of prompt optimization with LRMs and compare their performance with LLMs. In particular, we experimented with these models on a challenging task, i.e., end-to-end event extraction (EE), a structured prediction task of information extraction that requires identifying and classifying event triggers and arguments within text. EE poses unique challenges: models must follow schema constraints, handle coreference, and balance precision with recall, all of which demand nuanced reasoning. We evaluated four models, two LRMs (DeepSeek-R1, o1) and two LLMs (GPT-4.5, GPT-4o) as both task models and prompt optimizers within a Monte Carlo Tree Search (MCTS) framework (Wang et al., 2024b). This setup allows us to examine both task performance and prompt optimization quality under a consistent setting.

Our experimental results (Fig. 1) show that LRMs benefit substantially from prompt optimization, even when the training set for optimization is small, and they outperform LLMs in both task performance (as a task model) and optimization effectiveness (as a prompt optimizer). When used as optimizers, LRMs produce more precise prompts that align with human annotation heuristics, leading to faster convergence and lower variance in MCTS. Our error analysis further shows that these optimized prompts reduce common mistakes such as implicit trigger overgeneration or argument span drift. While DeepSeek-R1 as a prompt optimizer yields the most effective and concise prompts, prompt length alone is not predictive, i.e., different task models prefer different prompt styles. To test generality, we apply the same optimization framework to two tasks beyond EE, i.e., Geometric Shapes (Suzgun et al., 2022) and NCBI Disease NER (Doğan et al., 2014). In both, LRMs again show the largest gains, confirming that our findings extend beyond schema-based tasks.

## 2 RELATED WORKS

Prompt optimization has become an essential direction in adapting LLMs for downstream tasks without modifying their weights. For models with accessible internal states, such as open-source LLMs, prior work has explored soft prompt tuning (Li & Liang, 2021; Lester et al., 2021; Wang et al., 2023b; Hu et al., 2022) and gradient-based search methods that directly adjust prompt embeddings (Shin et al., 2020; Wen et al., 2023). Reinforcement learning has also been applied to optimize prompts through interaction-based feedback (Deng et al., 2022; Zhang et al., 2023).

However, these approaches are not applicable to closed-source LLMs accessed via APIs, where gradients and internal representations are unavailable. As such, research has focused on black-box, gradient-free techniques that rely on prompt perturbation and scoring. Many of these methods operate in an iterative loop: starting from an initial prompt, they generate variants, evaluate them on held-out examples, and retain the best one for the next round. Variants can be created through phrase-level edits (Prasad et al., 2023), back-translation (Xu et al., 2022), evolutionary operations (Guo et al., 2024; Fernando et al., 2024), or by prompting another LLM to rewrite the prompt based on model errors (Zhou et al., 2022; Pryzant et al., 2023; Srivastava et al., 2024; Wang et al., 2024b). Structured strategies such as Monte Carlo search (Zhou et al., 2022), Gibbs sampling (Xu et al., 2024), and beam search (Pryzant et al., 2023) have been explored to improve the efficiency of exploration.

More recent efforts have proposed structured prompt optimization. APE (Zhou et al., 2022) uses Monte Carlo Tree Search (MCTS) to explore the prompt space, while PromptBreeder (Fernando et al., 2024) and EvoPrompt (Guo et al., 2024) evolve prompts using feedback-driven mutation strategies. OPRO (Yang et al., 2024) employs mutation-based search guided by model performance. Other systems, such as PromptWizard (Agarwal et al., 2024) and Gödel Machine (Yin et al., 2025), incorporate self-evolving mechanisms in which the LLM iteratively generates, critiques, and refines its own prompts and examples.

While these approaches are promising, they have so far been applied exclusively to large, general-purpose LLMs. To the best of our knowledge, our work is the first to investigate prompt optimization for LRMs. Furthermore, we introduce and study this framework in the context of a structured prediction task, event extraction, which poses distinct challenges compared to typical mathematical or reasoning tasks explored in prior work (Zhou et al., 2022; Srivastava et al., 2024).

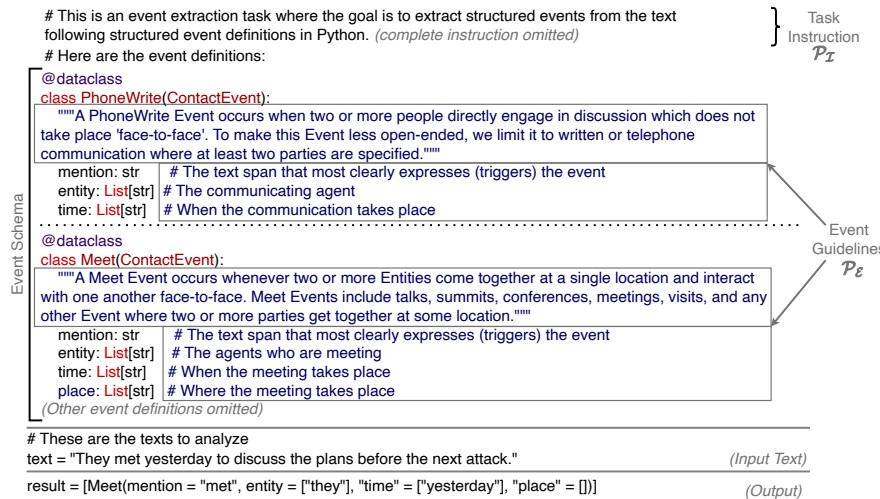

Figure 2: An example prompt for end-to-end Event Extraction (EE) used in our experiments, consisting of a task instruction and an event schema. The event schema contains information about the labels that are represented as Python classes and event guidelines defining both the event classes and the arguments. In prompt optimization, we refine both the task instruction and event guidelines (shown for two events; others omitted due to space limits) to generate more effective prompts for the task model.

## 3 METHODOLOGY

### 3.1 PROBLEM SETUP

Discrete prompt optimization aims to refine task-specific prompts for a task LLM $\mathcal{M}_{task}$ to improve its performance without modifying the model weights. In this study, we analyze whether LRMs benefit from prompt optimization in the context of *end-to-end EE*. The task consists of **trigger extraction**, which involves identifying event trigger spans and classifying their event types, and **argument extraction**, which requires identifying argument spans within the extracted event instance with a pre-defined role. To prompt a task model, $\mathcal{M}_{task}$, we adopted a Python code-based representation for both the input and the output of the model, which was shown to be effective by prior work (Wang et al., 2023a; Sainz et al., 2024; Li et al., 2023; 2024; Srivastava et al., 2025). As shown in Fig. 2, the initial prompt, $\mathcal{P}_0$ consists of two main parts: the task instruction and the event schema annotated by guidelines. **Task instruction** $\mathcal{P}_{\mathcal{I}}$ forms the initial segment of input to introduce the task and specify instructions such as the desired output format. The event schema contains information about the labels, such as event names and argument roles, that are represented as Python classes. The argument roles (e.g., time and place) are defined as attributes of event classes. All the events and arguments in a schema are annotated using human-written **event guidelines** $\mathcal{P}_{\mathcal{E}}$. The output is represented as a list of instances of the classes defined in the event schema. In this paper, we refine both $\mathcal{P}_{\mathcal{I}}$ and $\mathcal{P}_{\mathcal{E}}$ which is represented as the concatenation $\mathcal{P}_0 = [\mathcal{P}_{\mathcal{I}} || \mathcal{P}_{\mathcal{E}}]$, where $||$ represents the concatenation.

Given a training set $\mathcal{D}_{train} = \{(Q_i, A_i)\}_{i=1}^N$, where each $Q_i$ denotes an input text and $A_i$ its corresponding event instance, the objective of prompt optimization is to discover an **optimal prompt** $\mathcal{P}^*$ that maximizes a task-specific evaluation function $\mathcal{R}$, such as the F-score for EE. Event guidelines typically contain a combination of explicit schema constraints and implicit domain-specific rules that annotators follow during data labeling. However, not all of these rules are fully documented or easily translatable into a single static prompt. As a result, the initial prompt $\mathcal{P}_0$ may lack critical structural or interpretive cues required for high-quality extraction. We employ an optimizer LLM $\mathcal{M}_{opt}$ to refine $\mathcal{P}_0$ to discover such rules and constraints through strategic planning for superior, expert-level prompt optimization. Note that we do not modify the event schema defined by the original EE task, but only the human-written task instruction and the guidelines.

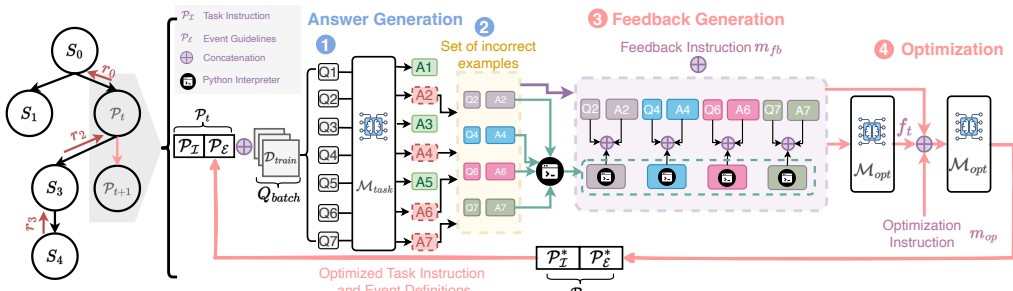

Figure 3: Overview of our prompt optimization framework. At each iteration, a zero-shot task LLM generates outputs, while a separate optimizer LLM analyzes the errors and updates the prompt, including task instructions and event guidelines, accordingly. This process continues over batches of training samples $\mathcal{D}_{train}$, and the final optimized prompt is evaluated on the development set to determine the node reward $r_t$.

## 3.2 PROMPT OPTIMIZATION FRAMEWORK

We frame prompt optimization as a discrete search over a large natural-language prompt space $\mathcal{S}$. Since $\mathcal{S}$ is too large for exhaustive search, we adopt Monte Carlo Tree Search (MCTS) to explore it efficiently, balancing exploration of new prompts with exploitation of promising ones, as in Wang et al. (2024b). We model the process as a Markov Decision Process (MDP) where each state $s_t$ is a prompt $\mathcal{P}_t$ and each action is formulated to make edits to the current prompt (e.g., adding constraints or clarifying rules).

Prompt optimization assumes a training set $\mathcal{D}_{train}$. As illustrated in Fig. 3, each MCTS node holds a prompt $\mathcal{P}_t$ and a batch of queries $Q_{batch}$ from the training set. In Step 1, the task model $\mathcal{M}_{task}$ is first employed to generate answers for queries in $Q_{batch}$. The incorrect outputs generated by the task model are then extracted and passed through a Python interpreter to identify issues such as parsing errors, missing event types, and invalid spans (Step 2). Following it, in Step 3, we prompt a prompt optimizer LLM $\mathcal{M}_{opt}$ with an instruction $m_{fb}$ to analyze the model errors and generate structured feedback $f_t$, including pinpointing unclear role definitions, proposing fixes, and summarizing recurring issues. In doing so, the generated feedback can be leveraged to produce targeted, actionable edits to improve clarity, coverage, and consistency of the task instruction and event guidelines. Next, in Step 4, $\mathcal{M}_{opt}$ is instructed by another instruction $m_{opt}$ to generate the updated prompt $\mathcal{P}_{t+1}$ in a single pass, based on the distribution $p_{\mathcal{M}_{opt}}(s_{t+1} \mid s_t, f_t, m_{opt})$. We also pass the history of previous prompts to discourage redundant edits. Only event types involved in the error batch are updated; others are inherited unchanged.

To evaluate each new prompt, we compute a reward $r_t = \mathcal{R}(s_t, f_t)$ based on averaged F1 scores across EE subtasks (TI, TC, AI, AC, described in Section 4.1) on a held-out development (dev) set after editing $\mathcal{P}_t$ with feedback $f_t$. The best prompt is selected based on dev-set performance. We provide additional details, the full algorithm, and the settings in Appendix A.

## 4 EXPERIMENTS

### 4.1 EXPERIMENTAL SETUP

**Dataset.** To evaluate the impact of prompt optimization on LRMs, we conduct experiments on the widely used ACE05 dataset (Doddington et al., 2004), a standard benchmark for EE that provides fine-grained event distinctions. We used the "split 1" preprocessed by Huang et al. (2024) and further processed it into the Python code format. The original ACE05 dataset includes 33 event types. However, our preliminary exploration found that including all 33 event types for prompt optimization could lead to overly long prompts, which both LLMs and LRMs cannot properly handle. To eliminate the impact of this confounding factor while assessing whether LRMs require and facilitate prompt optimization, we downsampled a subset of 10 event types in our experiments and left the issue of long-context processing as future work.

We utilize two smaller versions of ACE05 training set in our experiments as $\mathcal{D}_{train}$. To simulate low-resource conditions, we construct **ACE**$_{low}$ of 15 samples, where we select one instance per event type, prioritizing those with higher densities of event and argument annotations (i.e., training examples annotated with multiple event instances); the remaining samples are non-event instances. To examine the effect of scaling up the training size, we also construct a medium-scale dataset, **ACE**$_{med}$, comprising 120 examples—ten per event type—with the remaining being non-event instances. For both settings, we use a consistent development set of 100 examples randomly sampled from the ACE05 development set and focus our discussions about various task and optimizer models' performance on this set. For the full MCTS, we additionally report the model performance on a test set consisting of 250 examples randomly sampled from the ACE05 test set. Dataset statistics for **ACE**$_{low}$ and **ACE**$_{med}$ are summarized in Table 4 (Appendix A).

To test generalization beyond EE, we additionally include two tasks: **Geometric Shapes** (Suzgun et al., 2022), a symbolic reasoning benchmark, and **NCBI Disease NER** (Doğan et al., 2014), a biomedical named entity recognition task.

**Evaluation.** Following Huang et al. (2024), on EE, we evaluate models using four F1-based metrics: **(1) Trigger Identification (TI)**, which measures the correct extraction of trigger spans; **(2) Trigger Classification (TC)**, which additionally requires predicting the correct event type; **(3) Argument Identification (AI)**, which assesses the correct extraction of arguments and their association with the predicted trigger; and **(4) Argument Classification (AC)**, which further requires correct role labeling and serves as the most comprehensive measure of overall end-to-end EE performance. For analysis, we primarily report AC scores, which are widely regarded as a precise metric for evaluating both argument and trigger quality (Huang et al., 2024). Full results for all EE metrics are provided in Appendix B. For Geometric Shapes, we report test accuracy; for NCBI Disease NER, we report micro-F1 on strict disease spans.

**Experimental Settings and Baselines.** Our experiments involve two LRMs, DeepSeek-R1 and OpenAI-o1, and two general-purpose LLMs, GPT-4.5 and GPT-4o, used both as $\mathcal{M}_{opt}$ and $\mathcal{M}_{task}$. We conduct two sets of experiments. First, we evaluate all models trained on ACE$_{low}$ and ACE$_{med}$ using **shallow MCTS (depth 1)** to examine whether LRMs benefit from prompt optimization. We started with this design choice owing to its reduced complexity and computational costs. Next, we then perform **full MCTS (depth 5)** optimization on ACE$_{med}$ to investigate the deeper dynamics of optimization; ACE$_{low}$ is excluded from full-scale search due to its limited size. In each depth of rollout, we expand the parent node by three child expansions. For all our experiments, we report results only from the best-performing prompt nodes in each model's search trajectory. To reduce the inference cost, we followed Cheng et al. (2023) to employ "batch prompting" when querying $\mathcal{M}_{task}$ for answer generation (Step 1 in Fig. 3). Interestingly, we observed a performance gain than querying the task model for one question at a time. Due to policy restrictions, we were not allowed to access DeepSeek-R1 through API calls and thus deployed it locally on our own server. Because of our compute limit, we quantize DeepSeek-R1 to 2.5 bits using the UnSloth framework, which has shown minimal degradation in reasoning tasks even at lower precisions when rigorously benchmarked to 1.58 bits (Daniel Han & team, 2023). Additional details on batch prompting and hyperparameter configurations are provided in Appendix A.

## 4.2 EXPERIMENTAL RESULTS

Our main results are presented in Table 1. We discuss the following research questions (RQs).

**RQ1: Do LRMs benefit from prompt optimization in EE?** We first study whether the models can gain from prompt optimization by performing MCTS at depth 1. We observe consistent gains from prompt optimization across all models, with LRMs showing especially strong improvements over their non-optimized counterparts (around $+8\%$ on ACE$_{low}$ and $+23\%$ on ACE$_{med}$). LLMs also benefit from optimization, though to a lesser extent: GPT-4o and GPT-4.5 improve by around $+7\%$ and $+5\%$ on ACE$_{low}$, and by $+14\%$ and $+20\%$ on ACE$_{med}$, respectively. Overall, the performance gains from prompt optimization are more pronounced in LRMs than in LLMs.

Similarly, in cross-model comparisons using optimized prompts, LRMs remain highly competitive. On ACE$_{low}$, GPT-4.5 slightly outperforms o1 by about $+1\%$ AC but trails behind DeepSeek-R1 by

| $\mathcal{M}_{task}$ | Optimizer LLMs/LRMs ($\mathcal{M}_{opt}$) | | | | | #Output |
| | No Opt. | GPT-4o | GPT-4.5 | o1 | DS-R1 | Tokens |
|---|---|---|---|---|---|---|
| *MCTS at depth 1 trained on* $\text{ACE}_{low}$ *(Development Set)* | | | | | | |
| **GPT-4o** | 12.68 | 18.18 +5.50 | 16.67 +3.99 | 18.83 +6.15 | 20.15 +7.47 | 15.31 |
| **GPT-4.5** | 16.47 | 19.33 +2.86 | 16.47 00.00 | 19.32 +2.85 | 22.31 +5.84 | 24.57 |
| **o1** | 13.94 | 18.96 +5.02 | 18.57 +4.63 | 20.29 +6.35 | 21.92 +7.98 | 489.67 |
| **DS-R1** | 16.45 | 18.67 +2.22 | 18.57 +2.12 | 21.83 +5.38 | 24.66 +8.21 | 217.71 |
| *MCTS at depth 1 trained on* $\text{ACE}_{med}$ *(Development Set)* | | | | | | |
| **GPT-4o** | 12.68 | 22.32 +9.64 | **27.54** +14.86 | 26.30 +13.62 | 25.10 +12.42 | 17.31 |
| **GPT-4.5** | 16.47 | 29.63 +13.16 | 35.94 +19.47 | **36.51** +20.04 | 35.42 +18.95 | 28.75 |
| **o1** | 13.94 | 30.19 +16.25 | 36.67 +22.73 | **36.98** +23.04 | 36.96 +23.02 | 543.45 |
| **DS-R1** | 16.45 | 32.20 +15.75 | 37.14 +20.69 | 38.77 +22.32 | **40.00** +23.55 | 277.11 |
| *MCTS at depth 5 trained on* $\text{ACE}_{med}$ *(Development Set)* | | | | | | |
| **GPT-4o** | 12.68 | 28.04 +15.36 | 27.03 +14.35 | **28.57** +15.89 | 27.31 +14.63 | 17.55 |
| **GPT-4.5** | 16.47 | 32.35 +15.88 | 37.58 +21.11 | 36.22 +19.75 | **37.74** +21.27 | 32.65 |
| **o1** | 13.94 | 33.52 +19.58 | 37.78 +23.84 | 38.71 +24.77 | **39.81** +25.87 | 575.36 |
| **DS-R1** | 16.45 | 37.97 +21.52 | 38.40 +21.95 | 40.58 +24.13 | **44.26** +27.81 | 301.45 |
| *MCTS at depth 5 trained on* $\text{ACE}_{med}$ *(Test Set)* | | | | | | |
| **GPT-4o** | 13.33 | 26.94 +13.61 | 34.75 +21.42 | 30.59 +17.26 | **35.79** +22.46 | 27.00 |
| **GPT-4.5** | 14.29 | 27.31 +13.02 | 35.29 +21.00 | 36.59 +22.30 | **36.69** +22.40 | 35.56 |
| **o1** | 15.38 | 28.57 +13.19 | 36.73 +21.35 | **38.71** +23.33 | 37.86 +22.48 | 526.43 |
| **DS-R1** | 16.00 | 31.93 +15.93 | 41.98 +25.98 | 42.06 +26.06 | **43.75** +27.75 | 211.43 |

Table 1: AC (F1) scores using different $\mathcal{M}_{task}$ and $\mathcal{M}_{opt}$. #Output Tokens delineates the average number of output tokens from the task model, including reasoning and non-reasoning contents. The background shades indicate the choice of prompt optimizers, i.e., LRMs, LLMs, or no optimization. The best optimization result is in **bold** for each task model, while the highest relative improvement over the no-optimization baseline is **underlined**. We observe that LRMs not only benefit significantly from prompt optimization but also serve as strong prompt optimizers for other models.

roughly $+2\%$. On $\text{ACE}_{med}$, both LRMs outperform LLMs: o1 surpasses GPT-4.5 by $+0.5\%$ AC, and DeepSeek-R1 gains over approximately $+3.5\%$. These findings suggest that LRMs are not only more responsive to prompt optimization but also more capable in zero-shot EE settings. As we show in RQ2, this gap widens further when using the full-depth MCTS-based optimization strategy.

> **Insight 1**: Prompt optimization benefits all models, but LRMs gain more, no matter whether small and medium-sized training data is present.

**RQ2: How do LRMs perform under full-scale MCTS prompt optimization?**   To assess whether the advantages of LRMs persist at scale, we perform MCTS with a search depth of 5 across all models on $\text{ACE}_{med}$. While performance improves overall, we observe that the gains from full-scale optimization are incremental rather than dramatic when compared to the improvements observed with a single roll-out (i.e., depth 1) of MCTS. LRMs, however, still exhibit relatively stronger improvements. DeepSeek-R1, for instance, gains an additional $+4.26\%$ AC over its previous best ($40.00 \mapsto 44.26$). Similarly, o1 improves by $+2.83\%$ ($36.98 \mapsto 39.81$) when selecting the best optimizer across depths. In contrast, LLMs GPT-4.5 and GPT-4o show modest gains of only $+1.23\%$ ($36.51 \mapsto 37.74$) and $+1.03\%$ ($27.54 \mapsto 28.57$), respectively. Finally, we report each task model's performance on the test set, using the same best prompt searched on $\text{ACE}_{med}$. Consistently, we observed that LRMs benefit more from full MSTC prompt optimization than LLMs.

> **Insight 2**: Full-scale MCTS optimization yields non-dramatic gains over single-step optimization, but LRMs benefit more.

**RQ3: Do LRMs make better prompt optimizers?**   We evaluate each task model's performance when optimized using various LRMs and LLMs to investigate the quality of optimized prompts. In the low-resource setting ($\text{ACE}_{low}$, Depth 1), DeepSeek-R1 consistently outperforms all other optimizers across all task models. Compared to the best-performing LLM optimizer (GPT-4o), DeepSeek-R1 yields substantial gains: about $+2\%$ AC for optimizing GPT-4o ($18.18 \mapsto 20.15$), $+3\%$ for GPT-4.5 ($19.33 \mapsto 22.31$) and o1 ($18.96 \mapsto 21.92$), and $+6\%$ when optimizing itself ($18.67 \mapsto 24.66$). Notably,

| Examples of Task Instructions Optimized by Different Models |
| --- |
| **NO OPTIMIZATION** 
 Best Scores 
 TI - 39.29 
 TC - 33.93 
 AI - 16.47 
 AC - 16.47 

 # This is an event extraction task where the goal is to extract structured events from the text following structured event definitions in Python. (...) For each different event type, please output the extracted information from the text into a python list format (...) you should always output in a valid pydantic format: result = [EventName("mention" = "trigger", "arg1_key" = "arg1_span", ...), EventName("mention" = "trigger", "arg1_key" = "arg1_span", ...)]. (...) |
| **GPT-4o** 
 Best Scores 
 TI - 48.28 
 TC - 48.28 
 AI - 40.51 
 AC - 37.97 

 # This is an event extraction task where the goal is to extract structured events (...) 
 # Task Instructions: 1. For each different event type, output the extracted information from the text (...) 
 2. Structure the output in a valid Pydantic format: 'result = [EventName("mention" = "trigger", (...). 
 3. Adhere strictly to the described event descriptions (...). 
 4. Address special cases:- Appeals: Consider involved parties from prior related events as "prosecutor". 
 - Multiple roles may apply contextually; ensure complete information extraction. 
 - Implicit indications: If mentions like "filed", "concluded", etc.,(...) use context to clarify them.(...) |
| **GPT-4.5** 
 Best Scores 
 TI - 46.15 
 TC - 46.15 
 AI - 40.80 
 AC - 38.40 

 # This is an event extraction task for identifying and structuring events from text using Python-defined event classes. Each structured event consists of an event trigger word, an event type (...) 
 ## Instructions: 
 1. Span Extraction: 
 - Extract precise and concise spans for mentions and arguments, conveying the event or argument role clearly (...) 
 - Accurately identify roles using contextual cues, effectively resolving ambiguities while prioritizing explicit spans. If roles are unmentioned, leave them empty. (...) 
 3. Output Format: Please follow the Python-format(...) 
 4. Clarifications and Exceptions:- Note explicitly when roles have exceptions based on role definitions. 
 - Manage overlapping roles by following specific guidelines for span clarity and precision, (...) |
| **DEEPSEEK-R1** 
 Best Scores 
 TI - 56.60 
 TC - 56.60 
 AI - 44.26 
 AC - 44.26 

 # Event Extraction Task: Extract structured events from text using Python class definitions.(...): 
 1. Span Extraction:- Triggers: Minimal contiguous spans (verbs/nouns) directly expressing the event. Include both verbal and nominal forms ("death" = Die, "killings" = Die).(...) 
 - Arguments: - Remove articles ("a/an/the") and possessive pronouns EXCEPT when part of official names or temporal phrases ("The Hague", "the past year") 
 - Resolve pronouns AND POSSESSIVE NOUNS to named entities immediately using same-sentence antecedents ("airline's plan" → ["airline"]) 
 - Strip role/location/age descriptors from arguments ("Philadelphia lawyers" → "lawyers") (...) 
 - Keep FULL spans for crimes/money including sources/amounts ("stereo worth $1,750 from family") unless legal terms (...) 
 2. Special Handling:- Bankruptcy Triggers: "went bust" → EndOrg(...) 
 - Crime Spans: Retain full contextual clauses ("If convicted of killings...") without truncation 
 - Temporal Phrases: Keep original spans with articles when part of phrase ("the early 90's") 
 3. Output Rules: Always output in Python-format as (...) 
 4. Critical Exceptions:-(...) |
| **o1** 
 Best Scores 
 TI - 66.67 
 TC - 66.67 
 AI - 44.93 
 AC - 40.58 

 # This is an event extraction task where the goal is to extract structured events from the text following structured event definitions in Python. (...) 
 Keep argument references minimal by removing articles, possessives, or descriptive words unless they are crucial identifiers (e.g., "the retailer" → "retailer", "my uncle" → "uncle"). 
 # Important guidelines to address prior errors: 
 # 1. For each event trigger, use the single most relevant word (e.g., "bankruptcy" rather than "file for bankruptcy"). 
 # 2. For argument roles, also use minimal spans (e.g., "soldier" instead of "a soldier," "woman" instead of "a woman").(...) 
 # 4. For justice events (Sue, Appeal, Convict, SentenceAct, etc.): (...) 
 # 5. For transfers of money, watch for direct or indirect references to donations, (...) 
 # 6. Do not skip events implied by synonyms or indirect wording (e.g., "shutting down" → EndOrg, (...). 
 # 7. If there is more than one event in a single text, output each in a separate entry.(...) |

Table 2: Example task instructions optimized by different optimizers when $\mathcal{M}_{task}$ = DeepSeek-R1, which yielded the best performance for each optimizer. LRMs tend to emphasize actionable extraction rules and exception handling, while paying minimal attention to the task instruction and output format. Additionally, they often include illustrative examples (**in bold**) to facilitate span extraction.

among LLMs, GPT-4o performs better than GPT-4.5 as an optimizer in all task model settings, despite being weaker as a task model.

On the other hand, when a larger training set is available (ACE$_{med}$, Depth 1), we observe a shift. While LRM optimizers remain strong—achieving over +23% AC gain while optimizing themselves—GPT-4.5 shows a significant boost in effectiveness. It consistently outperforms GPT-4o as an optimizer and in some cases narrows the gap with LRMs, reaching 35.94 when optimizing itself and 36.67 when optimizing o1. Qualitatively, as shown in Table 2, DeepSeek-R1 enhances the optimized prompt $\mathcal{P}^*$ by adding precise extraction rules, such as removing articles ("a/an/the") and possessive pronouns (highlighted in blue), as well as critical exception cases for handling specific triggers (highlighted in pink). In contrast, o1 tends to generate a larger number of extraction rules, resulting in longer prompts. Both LRMs also include specific examples to guide extraction. LLMs, by comparison, focus more on task instructions and output formatting, typically generating shorter prompts with fewer examples. Among them, GPT-4.5 occasionally adds exception handling, though this behavior is less consistent than in LRMs. We provide additional examples of optimized task instruction and event guidelines in Appendix C, and include an additional analysis of the prompt quality in Section 5.

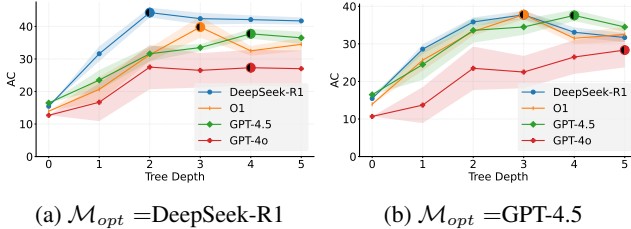

(a) $\mathcal{M}_{opt}$ =DeepSeek-R1      (b) $\mathcal{M}_{opt}$ =GPT-4.5

Figure 4: Convergence analysis of prompt optimization across different task models with two optimizers—DeepSeek-R1 (left) and GPT-4.5 (right). Task models converge faster with minimal variance when their prompts are optimized by LRMs.

> **Insight 3**: LRMs serve as highly effective optimizers, especially in low-resource settings where DeepSeek-R1 consistently outperforms all others as a prompt optimizer.

**RQ4: Can LRMs act as efficient and stable optimizers in prompt optimization?** In Fig. 4a, we observe that with DeepSeek-R1 as an optimizer, DeepSeek-R1 and GPT-4o demonstrate faster convergence compared to when GPT-4.5 is used as an optimizer (Fig. 4b), suggesting that it generates a higher quality of prompts. For DeepSeek-R1 and GPT-4.5 as task models, it also exhibits a smaller performance variance, which shows that R1 not only generates high-quality prompts but also does so reliably. In contrast, with GPT-4.5 as an optimizer, convergence tends to be slower. Under this setup, both LRMs reach their peak at depth 3, while GPT-4.5 and GPT-4o converge at depths 4 and 5, respectively. For GPT-4.5, the optimization process is visibly less stable than optimizing with DeepSeek-R1. Finally, we notice that most models begin to plateau, or slightly decline, beyond their optimal depth (marked using half-filled markers), reinforcing the presence of diminishing returns, where additional optimization yields increasingly smaller or no performance gains.

> **Insight 4**: DeepSeek-R1 (LRM) as an optimizer yields faster and more stable convergence than GPT-4.5 (LLM).

**RQ5: Do the optimization gains with LRMs generalize beyond schema-based tasks?** We further experimented on two tasks: Geometric Shapes and NCBI, and reported each task model's performance when we use the same model as an optimizer. As shown in Table 3, on both tasks, we observe that prompt optimization consistently improves all models. On Geometric Shapes, o1 and DeepSeek-R1 reach test accuracies of 77.80 and 78.40, outperforming GPT-4.5 (74.20) and GPT-4o (67.50). While GPT-4o achieves a larger rela-

| Model | No Opt. (Test) | Depth 1 (Dev) | Depth 5 (Dev) | Depth 5 (Test) |
|---|---|---|---|---|
| (a) Symbolic Reasoning — Geometric Shapes (Accuracy) | | | | |
| GPT-4o | 53.40 | 61.20 +7.80 | 68.67 +15.27 | 67.50 +14.10 |
| GPT-4.5 | 69.96 | 72.90 +2.94 | 75.33 +5.37 | 74.20 +4.24 |
| o1 | **70.07** | 73.50 +3.43 | 78.00 +7.93 | 77.80 +7.73 |
| DS-R1 | 69.67 | **73.80** +4.13 | **78.67** +9.00 | **78.40** +8.73 |
| (b) Biomedical IE — NCBI Disease NER (Micro-F1) | | | | |
| GPT-4o | 43.75 | 49.50 +5.75 | 54.37 +10.62 | 52.63 +8.88 |
| GPT-4.5 | **56.25** | 58.67 +2.42 | 65.56 +9.31 | 64.56 +8.31 |
| o1 | 53.13 | **66.46** +13.33 | **71.46** +18.33 | **70.15** +17.02 |
| DS-R1 | 54.20 | 66.00 +11.80 | 71.40 +17.20 | 69.96 +15.76 |

Table 3: Results on symbolic reasoning and biomedical NER tasks. Overall, LRMs benefit most from prompt optimization.

tive gain (+14.1), LRMs still achieve higher absolute performance. In NCBI, LRMs show strong gains and high final performance: o1 and DeepSeek-R1 improve by +17.0% and +15.8% F1, respectively, reaching 70.15 and 69.96, well above the LLM performance. These results mirror our findings on EE, reinforcing that LRMs not only serve as strong task models post-optimization but also generalize effectively as optimizers beyond schema-based tasks.

> **Insight 5**: Prompt optimization benefits transfer across tasks: LRMs gain benefit on both symbolic reasoning and biomedical NER.

## 5 FURTHER ANALYSIS

**Prompt Quality Across Optimizers** In addition to our qualitative analysis about Table 2 in RQ3, we also analyze the distribution of prompt effectiveness using a survival plot with DeepSeek-R1 as $\mathcal{M}_{task}$. The x-axis represents increasing AC thresholds, while the y-axis indicates the percentage of prompts that achieve at least that threshold. A higher survival curve indicates that an optimizer more consistently produces high-performing prompts. As shown in Fig. 5a, prompts optimized via

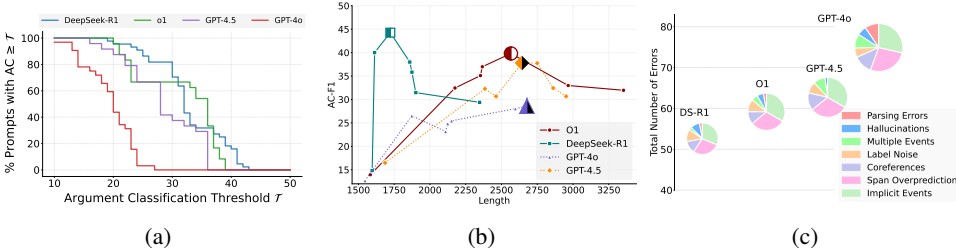

(a)  (b)  (c)

Figure 5: (a) A survival plot showing the % of prompts (y-axis) that achieve at least a given AC score (x-axis) for DeepSeek-R1 across different optimizers. (b) Prompt length vs. AC score across the best-performing full MCTS configuration for each task model on dev set. (c) Error categorization for DeepSeek-R1 as the task model with various optimizers.

DeepSeek-R1 exhibit the strongest survival curve, maintaining high-performance density even at stricter AC cutoffs ($\geq 35\%$ AC). In contrast, GPT-4o's curve decays rapidly, showing that while it occasionally generates effective prompts, its output quality is inconsistent. Interestingly, o1 and GPT-4.5 fall in between, with o1 slightly outperforming GPT-4.5 in the mid-range thresholds but trailing DeepSeek-R1 significantly at higher cutoffs. These trends reinforce our earlier findings: reasoning models are not only capable of producing better peak performance but also generate a greater density of usable prompts.

**Prompt Length vs. Task Model Performance**   To better understand how much instruction is needed for different task models to reach their peak performance, we analyze the relationship between prompt length and model accuracy across full MCTS search trees. For each model, we select its best-performing search trajectory (i.e., o1 as optimizer for GPT-4o and DeepSeek-R1 as optimizer for the other task models) and plot the corresponding full prompt lengths (including inherited definitions) against their AC scores in Fig. 5b. DeepSeek-R1 achieves its highest performance utilizing the shortest prompt ($\sim 1750$ tokens) in the search space, suggesting a preference for more concise task instructions. In contrast, both LLMs (GPT-4o and GPT-4.5) and the reasoning model o1 tend to rely on significantly longer prompts to achieve comparable accuracy.

**Error Analysis**   To better understand the types of errors introduced by different optimizers, we conduct a fine-grained analysis of all development examples where DeepSeek-R1 fails on prompts generated by different optimizers. As shown in Fig. 5c, LRMs notably reduce event-related errors, particularly those involving multiple or implicit events. Argument-related issues, such as coreference errors and span overprediction, are also slightly reduced. In some cases, all models produce non-parsable outputs or hallucinated argument spans. The remaining errors are primarily attributed to label noise in the dataset. We provide an example for each error category in Appendix B.

> **Insight 6**: LRM-optimized prompts are enriched with new extraction rules absent from the original task instruction, directly addressing frequent errors. DeepSeek-R1 achieves its highest performance using the shortest prompt.

## 6 CONCLUSION

We present the first systematic study of prompt optimization for LRMs, evaluating their roles as both task models and optimizers in a unified MCTS framework. On the structured task of event extraction, we find that LRMs benefit more from prompt optimization than LLMs and serve as stronger optimizers. They produce higher-quality prompts, converge faster, and generalize more reliably across models, highlighting their effectiveness in both prompt consumption and generation. Our error analysis further reveals that prompts optimized by LRMs reduce overprediction, hallucination, and parsing errors, contributing to more faithful and structured outputs. These trends generalize beyond event extraction: on Geometric Shapes and NCBI Disease NER, optimization improves all models, with LRMs outperforming LLMs when serving as their own optimizers. This strengthens our claim that LRMs both profit from and serve as strong agents for prompt optimization across diverse tasks.

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

## A ADDITIONAL DETAILS

### A.1 MORE IMPLEMENTATION DETAILS

To effectively optimize prompts for task-specific performance, we adopt a Monte Carlo Tree Search (MCTS) framework that iteratively explores and refines prompts based on model feedback and reward signals. The proposed algorithm, outlined in Algorithm 1, combines structured exploration with guided optimization by leveraging a task model, a feedback-generating optimizer, and a reward function. At each iteration, the algorithm performs selection, expansion, simulation, and back-propagation steps, progressively improving the prompt to maximize task performance across sampled batches.

---

**Algorithm 1** Algorithm for MCTS-based Prompt Optimization

**Inputs:**
 Initial prompt $s_0 = \mathcal{P}_0$, task model $\mathcal{M}_{task}$, optimizer $\mathcal{M}_{opt}$, reward function $\mathcal{R}$, batch size $k$, depth limit $L$, iterations $\tau$, exploration weight $c$

**Initialize:**
 State-action mapping $A : \mathcal{S} \mapsto \mathcal{F}$, children mapping $\text{ch} : \mathcal{S} \times \mathcal{F} \mapsto \mathcal{S}$, rewards $r : \mathcal{S} \times \mathcal{F} \mapsto \mathbb{R}$,
 Q-values $Q : \mathcal{S} \times \mathcal{F} \mapsto \mathbb{R}$, visit count $\mathcal{N} : \mathcal{S} \mapsto \mathbb{N}$

**for** $n \leftarrow 0, \ldots, \tau - 1$ **do**
 Sample batch $(Q_{batch}, A_{batch})$ from training data
 **for** $t \leftarrow 0, \ldots, L - 1$ **do**
  **if** $A(s_t)$ is not empty **then**                  ▷ selection

$$f_t \leftarrow \arg\max_{f \in A(s_t)} \left( Q(s_t, f) + c \cdot \sqrt{\frac{\ln \mathcal{N}(s_t)}{\mathcal{N}(\text{ch}(s_t, f))}} \right)$$

$$s_{t+1} \leftarrow \text{ch}(s_t, f_t), r_t \leftarrow r(s_t, f_t), \mathcal{N}(s_t) \leftarrow \mathcal{N}(s_t) + 1$$

  **else**                       ▷ expansion and simulation
   **(Step 1) Answer Gen:** $\hat{Q}_{batch} \sim \mathcal{M}_{task}(Q_{batch}, s_t)$
   **(Step 2) Error Extract:** Identify errors using interpreter on $\hat{A}_{batch}$
   **(Step 3) Feedback Gen:** $f_t \sim \mathcal{M}_{opt}(\text{feedback}|s_t, \text{errors})$
   **(Step 4) Prompt Update:** $s_{t+1} \sim \mathcal{M}_{opt}(s|s_t, f_t)$
   Update $A(s_t) \leftarrow \{f_t\}, \text{ch}(s_t, f_t) \leftarrow s_{t+1}, r(s_t, f_t) \leftarrow \mathcal{R}(\hat{A}_{batch}, A_{batch})$
   $r_t \leftarrow r(s_t, f_t), \mathcal{N}(s_t) \leftarrow \mathcal{N}(s_t) + 1$
  **end if**
  **if** $s_{t+1}$ is an early-stopping state **then**
   **break**
  **end if**
 **end for**
 $T \leftarrow$ number of steps
 **for** $t \leftarrow T - 1, \ldots, 0$ **do**                 ▷ back-propagation
  Update $Q(s_t, f_t)$ with rollout rewards $\{r_t, \ldots, r_L\}$
 **end for**
**end for**

---

| | Train $\mathbf{ACE}_{low}$ | Train $\mathbf{ACE}_{med}$ | Dev |
|---|---|---|---|
| **TransferMoney** | 3 | 13 | 29 |
| **Meet** | 2 | 15 | 13 |
| **PhoneWrite** | 1 | 11 | 1 |
| **SentenceAct** | 6 | 25 | 4 |
| **Appeal** | 2 | 16 | 4 |
| **Convict** | 5 | 11 | 5 |
| **Sue** | 3 | 13 | 8 |
| **EndOrg** | 1 | 11 | 1 |
| **Die** | 2 | 26 | 15 |
| **DeclareBankruptcy** | 1 | 11 | 1 |
| **None** | 5 | 20 | 30 |

Table 4: Data distribution for selected ETs.

## A.2 BATCH PROMPTING

Since querying LLMs individually for each input incurs substantial computational costs, a naïve approach that treats each input separately is inefficient. To mitigate this, we employ **batch prompting** (Cheng et al., 2023), which enables the combination of multiple queries into a single structured prompt. Given a batch of inputs $\{Q_1, Q_2, ..., Q_n\}$ that share the same task instruction $\mathcal{P}_\mathcal{I}$, batch prompting constructs a concatenated input string in the form $[\mathcal{P}_0||Q_1||Q_2||\ldots||Q_n]$. Each query is uniquely labeled (e.g., "text1") to maintain order and structure. The model processes this batch and generates a structured response in the form $[A_1||A_2||...||A_n]$, where each $A_i$ corresponds to the output for $Q_i$. These responses are parsed to extract individual predictions while preserving alignment. By reducing the number of API calls while maintaining high task accuracy, batch prompting improves efficiency, making large-scale prompt optimization feasible.

## A.3 PROMPT OPTIMIZATION AS A SEARCH PROBLEM

While batch prompting enhances efficiency, it does not inherently improve task performance. To address this, we formulate prompt optimization as a search problem over an expansive, intractable space of possible natural language prompts, denoted as $\mathcal{S}$. The objective is to discover an **optimal prompt** $\mathcal{P}^*$ that maximizes a task-specific evaluation function $\mathcal{R}$, such as the F-score for event extraction, formally defined as: $\mathcal{P}^* = \arg\max_{\mathcal{P}\in\mathcal{S}} \mathcal{R}(p_{\mathcal{M}_{task}}(A_{batch}|Q_{batch}, \mathcal{P}))$ where $Q_{batch}$ and $A_{batch}$ denote the batched queries and responses, respectively. Since this space is too large to exhaustively explore, we introduce a secondary LLM, $\mathcal{M}_{opt}$, which iteratively refines $\mathcal{P}_0$ based on errors observed in the output of $\mathcal{M}_{task}$. As shown in Fig. 3, this iterative refinement continues until a predefined stopping criterion is met, such as performance convergence or a fixed number of optimization steps. Once optimization concludes, the final optimized prompt $\mathcal{P}^*$ is used for inference on unseen test data.

## A.4 DATA SPLIT

We utilized two shorter versions of ACE05, $\text{ACE}_{low}$ and $\text{ACE}_{med}$. Their detailed descriptions are provided in Section 4.1. Table 4 presents the distribution of selected event types (ETs) across $\text{ACE}_{low}$, $\text{ACE}_{med}$, and the development (Dev) set. These subsets were curated to simulate both low-resource and medium-resource scenarios. Frequent ETs such as *SentenceAct* and *Die* contrast with rarer ones like *PhoneWrite* and *DeclareBankruptcy*, allowing for a diverse evaluation spectrum. The *None* class includes instances without any annotated events, preserving a realistic class distribution.

## A.5 META-PROMPTS FOR FEEDBACK ($m_{fb}$) AND OPTIMIZATION ($m_{opt}$)

**Feedback Collection Prompt.** Below we present the prompt $m_{fb}$ to collect structured feedback from $\mathcal{M}_{opt}$.

```
I am writing event guidelines and prompt (or task instructions) for a
    language model designed for an event extraction task.
```

```
My current prompt is:
<START>
{cur_prompt}
<END>

The event guideline in Python format is as following:
<START>
{event_definitions}
<END>

The task involves:
1. Extracting structured events (triggers, event type, arguments, and
    their roles) from the text.
2. Adhering to strict Python syntax for output (a Python list of event
    instances).
3. Handling all event definitions accurately, including mandatory roles
    and edge cases.

But this prompt gets the following examples wrong:
<START>
{example_string}
<END>

For each example, perform the following step-by-step analysis:
1. Error Type Classification: Identify the specific type(s) of error for
    each example (e.g., incorrect span extraction, missing roles,
    spurious arguments, format violations, etc.).
2. Root Cause Analysis:
    a. Did the current guideline fail to explain key extraction rules
        clearly?
    b. Are the instructions after '#' in the event definitions (
        guidelines) ambiguous, inconsistent, or insufficient?
    c. Were there ambiguities or overlaps in roles (e.g., 'agent' vs. '
        person') that caused confusion?
3. Example-Specific Recommendations:
    - Suggest precise changes to the guidelines (comments after '#' in
        event guidelines) to fix the errors for the given example.
    - Include explicit "what_to_do" and "what_not_to_do" instructions for
         ambiguous roles or edge cases.
    - Provide a simple example and counterexample to illustrate each
        guideline.
4. General Trends: Identify recurring issues in guidelines across all
    examples.

Expected Output:
1. For all the examples, summarize and list all actionable changes to
    improve the event definitions for all the classes, including:
    - Improved clarity for event/role definitions.
    - Enhanced handling of ambiguous or overlapping roles.
    - Guidelines for precise span extraction.

2. Provide an output pointing out the mistakes in the current guidelines
    and propose refinements for all the classes. Each refinement should
    include:
    - For an event, updated guidelines for "what_to_do" and "what_not_to_
        do."
    - Examples and counterexamples for each role.
```

**Task Instruction and Guidelines Optimization Prompt.**    Below we present the prompt $m_{opt}$ to optimize task instruction and event guidelines.

```
I am optimizing prompts for a language model designed for an event
    extraction task.
```

```
My current prompt (or task instructions) is:
<START>
{cur_prompt}
<END>

The event guideline in Python format is as following:
<START>
{event_definitions}
<END>

But this prompt gets the following examples wrong:
<START>
{example_string}
<END>

Based on these errors, the problems with the event guideline and the
    reasons are:
<START>
{feedback}
<END>

There are a list of former event guidelines including the current one,
    and each guideline is modified from its former prompts:
<START>
{trajectory_prompts}
<END>

Guidelines given to me for optimization of event classes:
1. Refine the prompt (or the task instructions) to address the issues
    mentioned previously. Focus on:
    - Clearer instructions for span extraction and role definitions along
        with any exceptions.
    - Handling ambiguous or overlapping roles effectively.
    - Strict adherence to Python-parsable output format.
2. Refine the guidelines for event definitions (the instructions after '#
    ') based on the identified mistakes. Ensure the refined guidelines
    addresses the concerns mentioned in the above.
3. Maintain backward compatibility: Ensure previously correct examples
    remain valid.
4. DO NOT change the ontology (Python classes). Instead, provide the
    refined guidelines in the format given at the end.
5. Ensure outputs follow these formats:
    - Optimized prompt (or the task instructions) wrapped with <START>
        and <END>.
    - Refined guidelines wrapped with <CLASS_START> and <CLASS_END>.

Output Requirements:
1. I have to provide the optimized prompt (or the task instructions) that
    evolves incrementally from the current one.
2. I also have to provide an output containing the fully optimized
    guidelines for each event definitions following the structure below:
class Event_Name(Parent_Event):
    \"\"\"
    # Updated guidelines here consulting the problems given to me
    \"\"\"
    mention: str # refined comments or extraction rules for event
    triggers. Include what/who can play the role with examples.
    {{role1}}: List # do the same for all roles including "mention",
    refining the comments after "#". Include what/who can play the role
    with examples and span extraction rule.

My response is:
```

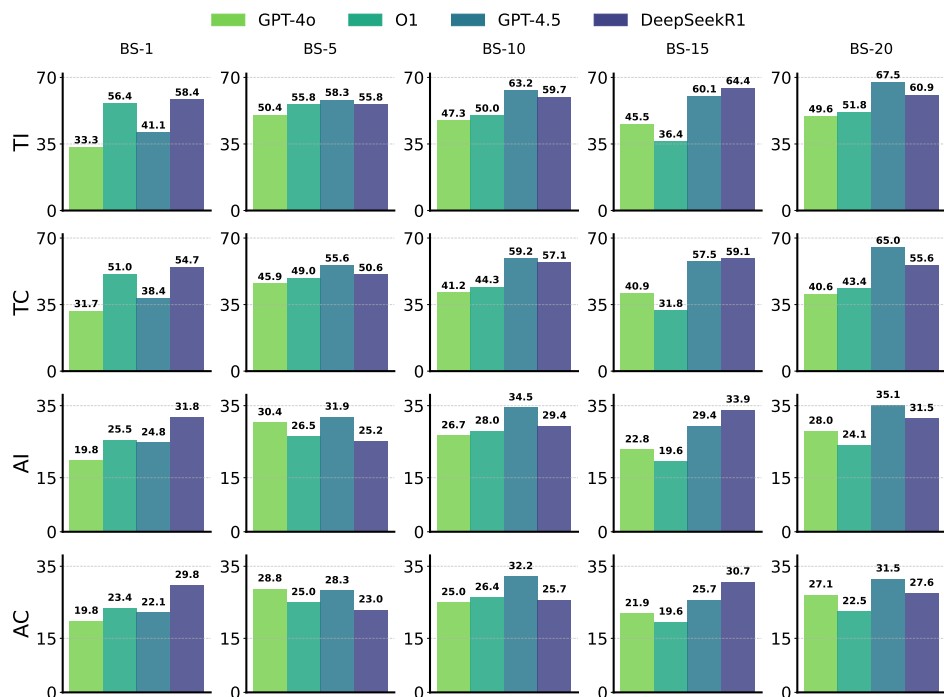

Figure 6: Batch-wise performance.

## A.6 Additional Hyperparameter and MCTS Configuration

Similar to Wang et al. (2024b), we provide the details of hyperparameters and Monte Carlo Tree Search (MCTS) configurations used in our experiments. For all runs, we fix the depth limit $L$ of the search tree to 5 and the number of MCTS iterations $\tau$ to 12, unless stated otherwise. The exploration-exploitation trade-off is controlled by the exploration weight $c$, which we set to 2.5 following prior work. The batch size $k$ for each rollout is set to 15.

We use greedy decoding for the task model $\mathcal{M}_{task}$ to simulate deterministic predictions, and temperature sampling with $T = 0.7$ for the optimizer model $\mathcal{M}_{opt}$ to promote diverse feedback generation. Early stopping in MCTS is triggered if a prompt leads to zero errors across two consecutive rollouts.

## A.7 Preliminary Experiments and Model Selection

Growing a full MCTS tree for prompt optimization can be computationally expensive, as noted in prior work Wang et al. (2024b). To establish a foundation before scaling up, we conducted initial experiments to analyze the impact of batch size on performance and computational efficiency. Since batch prompting reduces the number of API calls, we experimented with different batch sizes for constructing $Q_{batch}$ by varying the number of queries $Q_i$ and corresponding outputs $A_i$. However, we found that determining an optimal batch size for any LLM is highly model-dependent and lacks a universal heuristic (Fig. 6). Given this ambiguity, we set the batch size to 15, as it provides a straightforward 15-fold reduction in API calls while maintaining response quality. This choice ensured computational feasibility while allowing prompt optimization to operate effectively within our budget constraints. To further refine our experimental setup before scaling to a full MCTS search, we conducted an initial trial using a single iteration of MCTS. In this controlled setup, we instantiated a root node corresponding to the initial task prompt and generated three child nodes representing different prompt refinements. This limited exploration allowed us to assess the effect of prompt optimization for event extraction under different model settings.

## B  ADDITIONAL RESULTS AND ANALYSIS

### B.1  HOW DO OPTIMIZERS FOLLOW (OR IGNORE) FEEDBACK?

As mentioned in Section 5, optimizers exhibit different behaviors in how they apply feedback. For instance, we observed that in the majority of cases, DeepSeek-R1 refines only the event definitions that are explicitly mentioned in the feedback generated for the refinement of the task instruction and guidelines, leaving the remaining event definitions untouched. An example is shown in Figure 7, where DeepSeek-R1 reasons that the incorrect argument extraction for the `Attack` event likely stems from limitations of $\mathcal{M}_{task}$ rather than the guideline itself, and consequently refuses to modify it. In such cases, the unchanged definitions are inherited from the parent node.

To quantify this behavior, we measure the average number of *edited* guidelines and their average token length across all optimizers, under each model's best-performing configuration (based on AC score), in Figure 8. Notably, the token counts in this analysis differ from those in Figure 5b because we consider only the edited guidelines here—unedited ones are inherited from prior states—whereas the earlier analysis includes the full prompt content at each node. As shown in the figure, DeepSeek-R1 edits the fewest event types' guidelines (6.7 on average) and produces the shortest guidelines (approximately 1.5k tokens for guidelines edited in one optimization step), reflecting a more feedback-sensitive and token-efficient strategy. In contrast, GPT-o1 and GPT-4.5 modify nearly all ten guidelines (9.8 and 8.5 on average), regardless of feedback specificity, resulting in much longer outputs (2.9k and 2k tokens, respectively). While GPT-4o also appears restrained (7.6 edits on average), qualitative analysis suggests this is due to feedback overflow: when many suggestions are provided, GPT-4o often fails to address them all. These findings highlight DeepSeek-R1's more specific and efficient editing behavior, further reinforcing its strength as a prompt optimizer.

In this section, we present a comprehensive evaluation of various task models optimized through Monte Carlo Tree Search (MCTS) guided by different optimizer models. We analyze performance across multiple configurations, including varying dataset sizes (ACE$_{low}$, ACE$_{med}$, and ACE test set) and MCTS depths. Our analysis highlights how the interplay between task and optimizer models, as well as the depth of the optimization process, affects performance on trigger and argument prediction metrics.

*(truncated)* The model's prediction didn't include this, possibly because the coref wasn't resolved. The current guideline says pronouns require coref resolution before extraction. So if "his" is unresolved, it's omitted. But in the correct answer, "his" is resolved to Arafat. So the model might not have access to coref resolution, leading to omission. However, this is a model limitation, but the guideline says pronouns must be coref resolved. So maybe the guideline is clear, but the model isn't performing coref properly. Hence, I will skip the Attack event. *(truncated)*

Figure 7: Example reasoning when DeepSeek-R1 refuses to edit the Attack event.

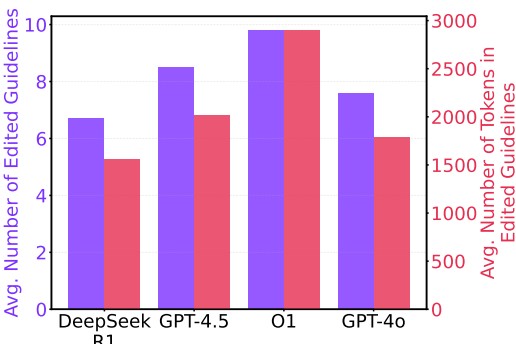

Figure 8: Average number of guidelines edited by each model and the average number of tokens in the edited guidelines for different optimizers when $\mathcal{M}_{task}$=DeepSeek-R1.

### B.2  FULL RESULTS

Table 5 compares the performance of four task models—DeepSeek-R1, o1, GPT-4.5, and GPT-4o—when optimized by different optimizer models across four key metrics: Trigger Identification (TI), Trigger Classification (TC), Argument Identification (AI), and Argument Classification (AC). Each row corresponds to a task model, and each column group corresponds to a specific optimizer guiding the prompt updates during MCTS. This layout allows us to analyze both the robustness of task models and the relative effectiveness of various optimizers under a shallow MCTS setup.

We further evaluate our method on the ACE$_{med}$ dataset using the same MCTS configuration with depth 1. Table 6 reports the performance of four task models under different optimizer models across the four standard evaluation metrics. Compared to ACE$_{low}$, this medium-resource setup

| Models | DeepSeek-R1 (Optimizer) | | | | o1 (Optimizer) | | | | GPT-4.5 (Optimizer) | | | | GPT-4o (Optimizer) | | | |
|---|---|---|---|---|---|---|---|---|---|---|---|---|---|---|---|---|
| | TI | TC | AI | AC | TI | TC | AI | AC | TI | TC | AI | AC | TI | TC | AI | AC |
| DeepSeek-R1 | 37.5 | 33.93 | 25.57 | 24.66 | 27.72 | 25.74 | 18.67 | 18.67 | 36.89 | 34.95 | 22.91 | 22.91 | 32.78 | 32.78 | 21.83 | 21.83 |
| o1 | 31.54 | 31.54 | 21.92 | 21.92 | 29.33 | 29.33 | 18.96 | 18.96 | 31.91 | 31.91 | 18.57 | 18.57 | 29.24 | 29.24 | 21.74 | 20.29 |
| GPT-4.5 | 36.04 | 34.23 | 23.14 | 22.31 | 34.78 | 33.04 | 20.07 | 19.33 | 31.37 | 31.37 | 19.32 | 19.32 | 30.29 | 30.29 | 20.97 | 20.19 |
| GPT-4o | 35.29 | 35.29 | 22.07 | 20.15 | 28.28 | 28.28 | 18.18 | 18.18 | 30.61 | 30.61 | 16.67 | 16.67 | 31.67 | 31.67 | 19.57 | 18.83 |

Table 5: Complete results of training on ACE$_{low}$ with MCTS depth 1 and tested on the dev set.

enables deeper insights into the generalizability and adaptability of both task and optimizer models. The results reveal notable variance in model-optimizer synergy, with certain combinations (e.g., o1 optimized by itself) yielding significantly stronger trigger performance, while others show more balanced gains across argument-level metrics.

| Models | DeepSeek-R1 (Optimizer) | | | | o1 (Optimizer) | | | | GPT-4.5 (Optimizer) | | | | GPT-4o (Optimizer) | | | |
|---|---|---|---|---|---|---|---|---|---|---|---|---|---|---|---|---|
| | TI | TC | AI | AC | TI | TC | AI | AC | TI | TC | AI | AC | TI | TC | AI | AC |
| DeepSeek-R1 | 63.16 | 63.16 | 40.00 | 40.00 | 65.45 | 65.45 | 32.2 | 32.2 | 56.25 | 56.25 | 37.14 | 37.14 | 62.7 | 62.7 | 40.06 | 38.77 |
| o1 | 78.95 | 78.95 | 39.13 | 36.96 | 54.78 | 54.78 | 33.96 | 30.19 | 59.26 | 59.26 | 36.67 | 36.67 | 57.14 | 57.14 | 36.98 | 36.98 |
| GPT-4.5 | 64.71 | 64.71 | 35.42 | 35.42 | 46.15 | 46.15 | 29.63 | 29.63 | 63.57 | 63.57 | 35.94 | 35.94 | 59.21 | 59.21 | 38.1 | 36.51 |
| GPT-4o | 30.00 | 30.00 | 25.88 | 25.1 | 28.57 | 28.57 | 22.32 | 22.32 | 34.55 | 34.55 | 27.54 | 27.54 | 29.38 | 29.38 | 26.99 | 26.3 |

Table 6: Complete results of training on ACE$_{med}$ with MCTS depth 1 and tested on the dev set.

We now report results on the ACE$_{med}$ dataset using a deeper MCTS configuration with depth 5. Table 7 summarizes the performance of each task model under four different optimizers. Compared to the shallower setup, this deeper search allows for more extensive prompt refinement, which can lead to either improved generalization or potential overfitting, depending on the optimizer-task model combination. Notably, certain models like o1 exhibit strong trigger-level performance when paired with GPT-4.5 as an optimizer, while others demonstrate more balanced gains across argument metrics. These results highlight the sensitivity of the optimization process to both the depth of MCTS and the choice of optimizer.

| Models | DeepSeek-R1 (Optimizer) | | | | o1 (Optimizer) | | | | GPT-4.5 (Optimizer) | | | | GPT-4o (Optimizer) | | | |
|---|---|---|---|---|---|---|---|---|---|---|---|---|---|---|---|---|
| | TI | TC | AI | AC | TI | TC | AI | AC | TI | TC | AI | AC | TI | TC | AI | AC |
| DeepSeek-R1 | 56.6 | 56.6 | 44.26 | 44.26 | 66.67 | 66.67 | 44.93 | 40.58 | 46.15 | 46.15 | 40.8 | 38.4 | 48.28 | 48.28 | 40.51 | 37.97 |
| o1 | 48.08 | 48.08 | 40.74 | 39.81 | 42.86 | 42.86 | 38.71 | 38.71 | 84.68 | 84.68 | 41.48 | 37.78 | 48.28 | 48.28 | 34.64 | 33.52 |
| GPT-4.5 | 45.68 | 45.68 | 38.36 | 37.74 | 51.24 | 51.24 | 36.22 | 36.22 | 59.26 | 59.26 | 36.24 | 37.58 | 41.18 | 41.18 | 32.35 | 32.35 |
| GPT-4o | 49.09 | 49.09 | 28.11 | 27.31 | 61.11 | 61.11 | 28.57 | 28.57 | 52.00 | 52.00 | 27.03 | 27.03 | 61.54 | 61.54 | 29.91 | 28.04 |

Table 7: Complete results of training on ACE$_{med}$ with MCTS depth 5 and tested on the dev set.

To assess the generalization capability of the optimized prompts, we evaluate all model-optimizer pairs on the ACE test set using an MCTS depth of 5. Table 8 presents the performance. This setup represents the final evaluation phase, where models are tested on unseen examples after undergoing deeper exploration-driven prompt optimization. Overall, the results show that performance trends remain consistent with those observed on the development set, though certain combinations—such as DeepSeek-R1 with itself as optimizer—demonstrate stronger stability, while others exhibit slight performance drops, especially in argument-level metrics. These observations reinforce the impact of both optimizer choice and MCTS depth on downstream generalization.

## B.3  ERROR CATEGORIES AND EXAMPLES

To better understand the limitations of our approach and the nature of model failures during prompt optimization, we conduct a qualitative error analysis by categorizing common mistakes observed in model outputs. Table 9 summarizes the key error categories encountered across multiple evaluation runs, along with representative examples and detailed descriptions. These categories—ranging from parsing issues and hallucinations to deeper linguistic challenges such as coreference and implicit

| Models | DeepSeek-R1 (Optimizer) | | | | o1 (Optimizer) | | | | GPT-4.5 (Optimizer) | | | | GPT-4o (Optimizer) | | | |
|---|---|---|---|---|---|---|---|---|---|---|---|---|---|---|---|---|
| | TI | TC | AI | AC | TI | TC | AI | AC | TI | TC | AI | AC | TI | TC | AI | AC |
| DeepSeek-R1 | 69.23 | 67.69 | 44.33 | 43.75 | 54.12 | 54.12 | 42.06 | 42.06 | 52.8 | 52.8 | 41.98 | 41.98 | 47.27 | 47.27 | 33.61 | 31.93 |
| o1 | 68.28 | 67.76 | 38.44 | 37.86 | 67.86 | 67.86 | 38.71 | 38.71 | 58.29 | 58.29 | 36.73 | 36.73 | 41.11 | 41.11 | 28.57 | 28.57 |
| GPT-4.5 | 68.31 | 68.31 | 38.44 | 36.69 | 64.71 | 64.71 | 39.02 | 36.59 | 56.45 | 56.45 | 35.29 | 35.29 | 49.09 | 49.09 | 28.11 | 27.31 |
| GPT-4o | 59.44 | 59.44 | 36.99 | 35.71 | 64.52 | 64.52 | 30.59 | 30.59 | 56.57 | 56.57 | 34.75 | 34.75 | 48.19 | 48.19 | 26.94 | 26.94 |

Table 8: Complete results of training on ACE$_{med}$ with MCTS depth 5 and tested on the test set.

event detection—highlight areas where models tend to struggle, particularly under batch prompting and complex event structures.

| Error Category | |
|---|---|
| **Parsing Errors** | **Description:** Parsing errors occur when the model's output is not in the expected format (e.g., JSON or structured list), often due to extra reasoning or verbose responses in batch prompts. These make the output unusable for evaluation pipelines. |
| | **Example:** Prompts that return extra text or commentary instead of a valid Python structure, causing non-parsable output. |
| **Hallucinations** | **Description:** Hallucinations occur when the model generates arguments or events that are not supported by the input. This usually happens due to biases learned during training or lexical overlaps with known labels. |
| | **Example: Text:** "Different parts of the strip saw conflicts today." → Model incorrectly predicts a 'Conflict' event based solely on the word "conflict". |
| **Multiple Events** | **Description:** Multiple event errors happen when the model detects only a single event in a sentence that contains multiple, usually defaulting to the most salient or final event. |
| | **Example: Text:** "...went home and his father-in-law killed him." → Model only predicts the 'Die' event, ignoring the 'Transport' event. |
| **Label Noise** | **Description:** Label noise refers to inconsistencies or ambiguities in the dataset annotations, such as differing treatment of coreferences or unclear event boundaries, which confuse both training and evaluation. |
| | **Example: Text:** "Our president has repeatedly... relied on a man... Hussein Kamel... leader of the Iraq arms program who defected..." → Label uses 'person=["leader"]'; model uses 'person=["Hussein Kamel"]'. |
| **Coreferences** | **Description:** Coreference errors arise when the model fails to resolve references like pronouns or role-based descriptors to their actual entities, leading to incorrect or incomplete argument spans. |
| | **Example: Text:** "...Hussein Kamel, leader of the Iraq arms program who defected..." → Label uses "leader"; model uses "Hussein Kamel", highlighting coreference resolution challenges. |
| **Span Overprediction** | **Description:** Span overprediction occurs when the model predicts more detailed argument spans than necessary, often including modifiers or descriptors not required by the task's minimal span rules. |
| | **Example: Text:** "Orders went out today to deploy 17,000 U.S. Army soldiers in the Persian Gulf region." → Label: "soldiers"; Prediction: "17,000 U.S. Army soldiers" – includes extra modifiers. |
| **Implicit Events** | **Description:** Implicit events are those not directly triggered by verbs but inferred through adjectives, nouns, or other context (e.g., "former"). These are often missed by models unless explicitly instructed. |
| | **Example: Text:** "...with former Congressman Tom Andrews..." → Trigger "former" implies 'EndPosition', but is often missed by models lacking rules for implicit event detection. |

Table 9: Description of error categories with examples.

## C   OPTIMIZED TASK INSTRUCTION AND GUIDELINES

In this section, we present fully optimized task instruction and event guidelines generated by DeepSeek-R1, o1, GPT-4.5, and GPT-4o.

### C.1   EXAMPLE OF OPTIMAL TASK INSTRUCTION AND EVENT GUIDELINES GENERATED BY DEEPSEEK-R1

```
# Event Extraction Task: Extract structured events from text using Python
    class definitions. Follow these rules:

1. **Span Extraction**:
   - **Triggers**: Minimal contiguous spans (verbs/nouns) directly
     expressing the event. Include both verbal and nominal forms ("
     death" = Die, "killings" = Die). Add new triggers like "converge"
     for Meet and "is_no_more" for EndOrg
   - **Arguments**:
     - Remove articles ("a/an/the") and possessive pronouns EXCEPT when
        part of official names or temporal phrases ("The_Hague", "the_
        past_year")
     - Resolve pronouns AND POSSESSIVE NOUNS to named entities **
        immediately** using same-sentence antecedents ("airline's_plan"
        → ["airline"])
     - Strip role/location/age descriptors from arguments ("Philadelphia_
        lawyers" → "lawyers") unless part of multi-word crime
     - Keep FULL spans for crimes/money including sources/amounts ("
        stereo_worth_$1,750_from_family") unless legal terms
     - Detect beneficiaries via ownership markers ("for_X's_project"),
        direct "to_X" transfers go to recipient

2. **Special Handling**:
   - **Bankruptcy Triggers**: "went_bust" → EndOrg unless explicit
     bankruptcy context
   - **Meet Entities**: Include ALL resolvable participants (subject +
     object)
   - **Crime Spans**: Retain full contextual clauses ("If_convicted_of_
     killings...") without truncation
   - **Temporal Phrases**: Keep original spans with articles when part of
      phrase ("the_early_90's")

3. **Output Rules**:
   - Always output in Python-format as [EventName("mention" = "trigger",
     "arg1_key" = "arg1_span", ...), EventName("mention" = "trigger", "
     arg1_key" = "arg1_span", ...)]
   - Include ALL role fields with empty lists where applicable
   - Output separate events for each trigger (no merging) even for
     identical event types
   - Strict pydantic syntax: [EventName(mention="span", role=["span"],
     ...)]
   - Preserve original casing for locations unless explicitly proper
     nouns

4. **Critical Exceptions**:
   - **EndOrg Triggers**: Add "collapse", "drive_out", "went_bust" with
     explicit org mentions
   - **Appeal Roles**: defendant = opposing party (state), prosecutor =
     appellant
   - **TransferMoney**: "for_X" → recipient unless ownership marker ("for
     _X's_Y" → beneficiary)
   - **PhoneWrite Entities**: Strip ALL role descriptors ("Secretary_
     Powell" → ["Powell"])
```

```
# Here are the event definitions:

class Convict(JusticeEvent):
    """Extract convictions where entity is found guilty of crime.
    Key Updates:
    - crime: Retain FULL spans including amounts/sources ("received
        stereo worth $1,750 from family")

    Example: "convicted of taking bribes worth $1M" → crime=["taking
        bribes worth $1M"]
    Counterexample: Truncating to ["taking bribes"] → error
    """
    mention: str  # Triggers: "convicted", "conviction"
    defendant: List[str]  # ["Vang"] (resolved pronouns, strip
        descriptors)
    adjudicator: List[str]  # ["court"] (official names only)
    crime: List[str]  # Full offense span without legal terms
    time: List[str]  # ["last Wednesday"] (exact temporal phrases)
    place: List[str]  # ["Minnesota"] (geopolitical entities from context
        )

class TransferMoney(TransactionEvent):
    """Money transfers without goods exchange.
    Key Updates:
    - recipient: Direct receiver ("to X" OR "for X" if X is endpoint)
    - beneficiary: Only for ownership ("for X's project") or indirect
        benefit

    Example: "donated $5 for Tim Kaine" → recipient=["Tim Kaine"]
    Example: "funds for Kaine's campaign" → beneficiary=["Kaine"]
    """
    mention: str  # Triggers: "provided money", "donation"
    giver: List[str]  # ["foundation"] (strip descriptors)
    recipient: List[str]  # ["charity"] (direct receiver from "to/for X")
    beneficiary: List[str]  # ["Suha"] (from ownership markers)
    money: List[str]  # ["$15M"] (keep symbols/approximations)
    time: List[str]  # ["two years"] (full temporal span)
    place: List[str]  # ["Swiss"] (origin locations, strip prepositions)

class Meet(ContactEvent):
    """Face-to-face interactions.
    Key Updates:
    - entity: Include ALL resolvable participants (subject + object)

    Example: "Annan met Al-Douri" → entity=["Annan", "Al-Douri"]
    Counterexample: Omitting subject → error
    """
    mention: str  # Triggers: "meet", "summit", "talks"
    entity: List[str]  # ["delegates"] (all participants)
    time: List[str]  # ["today"] (exact temporal span)
    place: List[str]  # ["Dallas"] (resolved location noun)

class PhoneWrite(ContactEvent):
    """Non face-to-face communication.
    Key Updates:
    - entity: Strip ALL role descriptors unless part of compound name

    Example: "e-mail from Secretary Powell" → entity=["Powell"]
    Counterexample: Retaining "Secretary" → error
    """
    mention: str  # Triggers: "called", "e-mail" with transmission
        context
    entity: List[str]  # ["we", "them"] (bare names, resolved pronouns)
    time: List[str]  # ["during meeting"] (exact time phrase)
```

```
     place: List[str]  # ["office"] (specific location if present)

class DeclareBankruptcy(BusinessEvent):
    """Formal bankruptcy declarations.
    Key Rules:
    - entity: Resolve org pronouns AND possessive nouns ("airline's
        bankruptcy" → ["airline"])
    - Triggers: "bankruptcy", "Chapter 11" (exclude "collapse"/"went bust
        " without explicit bankruptcy context)

    Example: "airline's bankruptcy filing" → mention="bankruptcy", org=["
        airline"]
    Counterexample: "near-collapse" → EndOrg
    """
    mention: str  # Triggers indicating financial collapse: "bankruptcy",
         "Chapter 11"
    entity: List[str]  # ["Enron Corp"] (resolved orgs from pronouns/
        possessives in same sentence)
    time: List[str]  # ["2003"] (declaration time phrase)
    place: List[str]  # ["Texas"] (jurisdiction noun if specified)

class EndOrg(BusinessEvent):
    """Organization termination events.
    Key Rules:
    - Triggers: "ceased", "is no more", "collapse", "drive out", "went
        bust"
    - org: Require explicit organizational mention ("casinos" in "casinos
        faced collapse")

    Example: "company went bust" → mention="went bust", org=["company"]
    Counterexample: "facing collapse" (no explicit org) → ignore
    """
    mention: str  # Triggers must indicate actual termination
    org: List[str]  # ["plant"] (direct object or possessive noun)
    time: List[str]  # ["the past year"] (with articles when part of
        phrase)
    place: List[str]  # ["Eugene"] (specific location noun)

class Die(LifeEvent):
    """Death events.
    Key Updates:
    - mention: Include nominal forms ("killings", "casualties") as valid
        triggers

    Example: "massacre casualties" → mention="casualties"
    Counterexample: "death penalty" → ignore
    """
    mention: str  # Triggers: "died", "killings", "casualties"
    agent: List[str]  # ["shooter"] (intentional actors only)
    victim: List[str]  # ["patient"] (without quantifiers/possessives)
    instrument: List[str]  # ["knife"] (specific tools/weapons)
    time: List[str]  # ["last night"] (exact span)
    place: List[str]  # ["hospital"] (death location noun)

class SentenceAct(JusticeEvent):
    """Punishment issuance events.
    Key Updates:
    - crime: Retain original crime from conditional clauses ("If
        convicted of killings..." → ["killings"])

    Example: "faces life for fraud" → crime=["fraud"]
    Counterexample: "could face penalty" → ignore
    """
```

```
        mention: str  # Triggers: "sentenced", "faces". Must reference actual
            punishment
        defendant: List[str]  # ["activist"] (strip role descriptors)
        adjudicator: List[str]  # ["jury"] (bare roles unless official title)
        crime: List[str]  # ["illegally attending meeting"] (full contextual
            span)
        sentence: List[str]  # ["life in prison"] (exact punishment phrase)
        time: List[str]  # ["Thursday"] (exact temporal expression)
        place: List[str]  # ["district court"] (decision location noun)

class Sue(JusticeEvent):
    """Legal action initiations.
    Key Updates:
    - adjudicator: Include "judge" if overseeing case approval ("approved
        by judge" → ["judge"])

    Example: "suit against Gateway approved by judge" → adjudicator=["
        judge"]
    Counterexample: "lawsuit documents" → adjudicator=[]
    """
        mention: str  # Triggers: "suit", "lawsuit". Must reference legal
            filing
        plaintiff: List[str]  # ["patients"] (strip locations/roles unless
            critical)
        defendant: List[str]  # ["Gateway"] (explicitly sued entities)
        adjudicator: List[str]  # ["judge"] (if directly involved)
        crime: List[str]  # ["malpractice"] (explicit offense without legal
            terms)
        time: List[str]  # ["last month"] (keep articles in temporal phrases)
        place: List[str]  # ["South Florida"] (specific noun phrases)

class Appeal(JusticeEvent):
    """Court decision appeals.
    Key Updates:
    - defendant: Opposing party (state/prosecution), NOT appellant
    - prosecutor: Entity filing appeal (resolved from subject/pronouns)

    Example: "appeal by Anwar against conviction" → prosecutor=["Anwar"],
        defendant=[]
    Counterexample: Assigning appellant as defendant → error
    """
        mention: str  # Triggers: "appeal", "appeals"
        defendant: List[str]  # ["state"] (opposing party in original case)
        prosecutor: List[str]  # ["Pasko"] (appellant, bare name without
            roles)
        adjudicator: List[str]  # ["court"] (original court name)
        crime: List[str]  # ["espionage"] (original charge)
        time: List[str]  # ["last week"] (exact temporal phrase)
        place: List[str]  # ["Malaysia"] (country from court description)
```

## C.2 EXAMPLE OF OPTIMAL TASK INSTRUCTION AND EVENT GUIDELINES GENERATED BY O1

```
# This is an event extraction task where the goal is to extract
    structured events from the text following structured event
    definitions in Python.
# A structured event contains:
#   (1) an event trigger word (mention) -- always use the minimal lexical
     span (e.g., "appeal" rather than "filed an appeal"),
#   (2) an event type, and
#   (3) the arguments participating in the event (with their roles).

# Keep argument references minimal by removing articles, possessives, or
    descriptive words unless they are crucial identifiers (e.g., "the
    retailer" -> "retailer", "my uncle" -> "uncle").
```

```
# Important guidelines to address prior errors:
#    1. For each event trigger, use the single most relevant word (e.g., "
     bankruptcy" rather than "file for bankruptcy").
#    2. For argument roles, also use minimal spans (e.g., "soldier"
     instead of "a soldier," "woman" instead of "a woman").
#    3. Output a separate event for each distinct trigger or implied event
      (e.g., a conviction and a subsequent sentencing should be two events
     ).
#    4. For justice events (Sue, Appeal, Convict, SentenceAct, etc.):
#         - "defendant" is the party or entity accused or found guilty.
#         - "plaintiff" or "prosecutor" is the party initiating legal
     action or bringing an appeal. If the text does not specify who is
     accused, leave "defendant" empty.
#         - If the text refers to a punishment or sentencing (e.g., "faces
      the death penalty"), include a separate SentenceAct event referencing
      the same "defendant."
#    5. For transfers of money, watch for direct or indirect references to
      donations, funding, or contributions and label them as TransferMoney
      events.
#    6. Do not skip events implied by synonyms or indirect wording (e.g.,
     "shutting down" → EndOrg, "emerged from bankruptcy" →
     DeclareBankruptcy).
#    7. If there is more than one event in a single text, output each in a
      separate entry.
#    8. Always produce valid Python list format exactly as:
#         result = [
#            EventName("mention" = "trigger", "role1" = [...], "role2" =
     [...], ...),
#            EventName("mention" = "trigger", "role1" = [...], "role2" =
     [...], ...),
#         ]
#    9. Do not output anything else except this parsable Python structured
      format (no extra text or explanation).

# The event class definitions remain the same, but refer to the following
     refined docstrings for usage examples, minimal spans, and role
     clarifications.

# Here are the event definitions:

class Convict(JusticeEvent):
    """
    A Convict Event occurs whenever a Try Event ends with a successful
        prosecution of the Defendant.
    In other words, a Person, Organization or GPE Entity is convicted
        whenever that Entity has been
    found guilty of a Crime.

    Refined Guidelines:
      • mention: Use the minimal trigger word referring to the conviction
          (e.g., "guilty", "convicted").
      • defendant: The entity/ies found guilty. Remove articles or
          possessives ("the man" → "man").
      • adjudicator: The court or judge that issued the guilty verdict,
          if explicitly given.
      • crime: The wrongdoing for which the defendant was found guilty (e
          .g., "murdering X").
      • time: Any explicit time references (e.g., "last week").
      • place: Any explicit location references (e.g., "in Boston").

    What to do:
        - Include "crime" if stated: e.g., "convicted of murdering his wife
            " → crime=["murdering his wife"].
```

```
        - Keep the defendant arg minimal: "Scott Peterson" → ["Scott
            Peterson"], not ["Mr. Scott Peterson"].

    What not to do:
        - Do not guess or infer the crime if not stated.
        - Do not prepend articles or descriptive words (e.g., "the
            defendant" → "defendant" if used generically).

    Example:
      Text: "John was found guilty of fraud."
      → Convict(mention='guilty', defendant=['John'], crime=['fraud'],
          time=[], place=[])
    """
    mention: str  # minimal word expressing the conviction event
    defendant: List[str]  # who is found guilty
    adjudicator: List[str]  # the judge or court, if stated
    crime: List[str]  # the wrongdoing for which the defendant is
        convicted
    time: List[str]  # when the conviction takes place
    place: List[str]  # where the conviction takes place

class TransferMoney(TransactionEvent):
    """
    TransferMoney Events refer to giving, receiving, borrowing, or
        lending money
    when not purchasing goods or services in return.

    Refined Guidelines:
        ● mention: Single word that triggers the transfer event (e.g., "
            donated", "loaned").
        ● giver: The agent who provides funds. Remove determiners ("the", "
            a") unless part of a name.
        ● recipient: The agent who receives the funds.
        ● beneficiary: Any additional agent that benefits, if separate from
             recipient.
        ● money: The amount of funds (if any mention like "$3,000", "large
            sum").
        ● time: When the event takes place (e.g., "today", "last year").
        ● place: Where the transaction or transfer occurs.

    What to do:
        - Label intangible references (e.g., "contributed", "had
            contributors") as TransferMoney if it implies funds.
        - Use minimal references for all money roles.

    What not to do:
        - Do not label intangible help (e.g., "emotional support") as
            TransferMoney.
        - Avoid listing indefinite articles or extraneous descriptors in
            the agent spans.

    Example:
      Text: "He donated $5,000 to Red Cross last week."
      → TransferMoney(mention='donated', giver=['He'], recipient=['Red
          Cross'], money=['$5,000'], time=['last week'], place=[])
    """
    mention: str  # minimal word triggering the money transfer
    giver: List[str]  # who provides the money
    recipient: List[str]  # who receives the money
    beneficiary: List[str]  # who additionally benefits, if any
    money: List[str]  # the sum or amount
    time: List[str]  # when the transfer happens
    place: List[str]  # where the transfer event occurs
```

```
class Meet(ContactEvent):
    """
    A Meet Event occurs when two or more Entities come together face-to-
        face
    at a single location and interact with one another.

    Refined Guidelines:
        • mention: The single best word for the meeting (e.g., "met", "
            summit", "conference").
        • entity: All participants, stripped of articles or descriptors. If
             multiple, list them all.
        • time: Any temporal phrase referencing when the event took place.
        • place: The location of the meeting.

    What to do:
        - Use triggers for in-person gatherings (e.g., "met", "conference",
             "summit").
        - Keep participant references minimal: "President", "Vice-President
            " instead of "the US President".

    What not to do:
        - Do not treat phone calls or written communication as Meet (use
            PhoneWrite).

    Example:
        Text: "The leaders met in Paris yesterday."
        → Meet(mention='met', entity=['leaders'], time=['yesterday'], place
            =['Paris'])
    """
    mention: str  # minimal word or short phrase for the meeting
    entity: List[str]  # who met face-to-face
    time: List[str]  # when the meeting happened
    place: List[str]  # where the meeting occurred

class PhoneWrite(ContactEvent):
    """
    A PhoneWrite Event occurs when two or more people communicate
    without meeting face-to-face. This includes phone calls, email,
        texting, etc.

    Refined Guidelines:
        • mention: The minimal expression of communication (e.g., "called",
             "emailed", "texted").
        • entity: The agents communicating. Strip out articles, determiners
            , or extra descriptors.
        • time: When the communication took place (e.g., "this morning", "
            yesterday").

    What to do:
        - Common triggers: "phoned", "emailed", "talked by phone", "texted
            ", "messaged".
        - Keep roles minimal (e.g., entity=['John', 'Mary']).

    What not to do:
        - Do not mark in-person discussions as PhoneWrite (use Meet).

    Example:
        Text: "They emailed each other last night."
        → PhoneWrite(mention='emailed', entity=['They'], time=['last night
            '])
    """
    mention: str  # minimal communication trigger
    entity: List[str]  # communicating parties
```

```
      time: List[str]  # when the communication happened

class DeclareBankruptcy(BusinessEvent):
    """
    A DeclareBankruptcy Event occurs whenever an Entity officially seeks
        legal protection
    from debt collection due to severe financial distress.

    Refined Guidelines:
      ● mention: Short trigger related to bankruptcy (e.g., "bankruptcy",
          "filed", "declared").
      ● org: The organization or person who declares bankruptcy. Remove "
          the", "my", etc.
      ● time: When the bankruptcy is declared (e.g., "in 2003", "today").
      ● place: Where the declaration is made, if mentioned (e.g., "in
          court", "in New York").

    What to do:
      - Recognize synonyms or indirect references like "emerged from
          bankruptcy" or "bankruptcy protection" as triggers.

    What not to do:
      - Do not guess an org if not specified.

    Example:
      Text: "My uncle declared bankruptcy in 2003."
      → DeclareBankruptcy(mention='bankruptcy', org=['uncle'], time
          =['2003'], place=[])
    """
    mention: str  # minimal expression for bankruptcy
    org: List[str]  # the party declaring bankruptcy
    time: List[str]  # when the declaration takes place
    place: List[str]  # where it is declared

class EndOrg(BusinessEvent):
    """
    An EndOrg Event occurs when an Organization ceases to exist or
    "goes out of business."

    Refined Guidelines:
      ● mention: Minimal trigger (e.g., "shutting down", "closing").
      ● org: The organization or sub-unit that ends. E.g., "plant", "
          branch".
      ● time: When this closure or end is stated to happen.
      ● place: Where the organization is located or ended.

    What to do:
      - Consider references such as "closing its plant" → "plant" in org.
      - Identify synonyms like "shutting down," "ceasing operations."

    What not to do:
      - Do not skip it if the text explicitly says the org ended.

    Example:
      Text: "Hewlett Packard is shutting down its plant in Eugene."
      → EndOrg(mention='shutting down', org=['plant'], time=[], place=['
          Eugene'])
    """
    mention: str  # minimal expression for the organizational end
    org: List[str]  # the ended organization
    time: List[str]  # when the end occurs
    place: List[str]  # where this event happens
```

```
class Die(LifeEvent):
    """
    A Die Event occurs whenever a Person loses their life, whether
        accidental,
    intentional, or self-inflicted.

    Refined Guidelines:
      • mention: The short trigger referencing the death (e.g., "killed",
          "died", "murdered").
      • agent: The killer or cause if identified (e.g., "gunman", "regime
          ")-remove articles.
      • victim: Who died, again with minimal references (e.g., "soldier"
          instead of "a soldier").
      • instrument: The device or method used, if any (e.g., "gun", "bomb
          ").
      • time: When the death occurred.
      • place: Where it took place.

    What to do:
      - Create separate Die events for each death trigger in the text.
      - If the text references homicide: agent is the killer, victim is
          the deceased.

    What not to do:
      - Do not combine multiple victims into one string if they appear as
          separate triggers.

    Example:
      Text: "He killed the soldier in Iraq."
      → Die(mention='killed', agent=['He'], victim=['soldier'],
          instrument=[], time=[], place=['Iraq'])
    """
    mention: str  # minimal word referencing the death
    agent: List[str]  # optional killer or cause
    victim: List[str]  # who died
    instrument: List[str]  # how they were killed (weapon, etc.)
    time: List[str]  # when the death happened
    place: List[str]  # where the death happened

class SentenceAct(JusticeEvent):
    """
    A SentenceAct Event occurs whenever a punishment for the Defendant is
        issued,
    e.g., a prison term or another legal penalty.

    Refined Guidelines:
      • mention: A trigger referencing sentencing or punishment (e.g., "
          sentenced", "faces [penalty]").
      • defendant: The same party convicted or found guilty, if known.
      • adjudicator: The entity delivering the sentence, if stated (e.g.,
          "judge", "court").
      • crime: The wrongdoing for which the defendant is sentenced (e.g.,
          "murder", "embezzlement").
      • sentence: The specific punishment (e.g., "death penalty", "life
          in prison").
      • time: When the sentencing occurs.
      • place: Where the sentencing occurs.

    What to do:
      - Look for words like "faces the death penalty," "was sentenced to
          ten years."

    What not to do:
```

```
                - Do not omit a SentenceAct if there's explicit mention of
                    punishment.

        Example:
            Text: "He now faces the death penalty for murdering his wife."
            → SentenceAct(mention='faces', defendant=['He'], crime=['murdering
                his wife'], sentence=['death penalty'], time=[], place=[])
        """
        mention: str  # minimal expression for the sentencing event
        defendant: List[str]  # who is sentenced
        adjudicator: List[str]  # judge or court
        crime: List[str]  # the wrongdoing or offense
        sentence: List[str]  # the punishment
        time: List[str]  # when the sentencing happens
        place: List[str]  # where it happens

class Sue(JusticeEvent):
        """
        A Sue Event occurs whenever a court proceeding is initiated to
            determine
        the liability of a Person, Organization, or GPE.

        Refined Guidelines:
            ● mention: The minimal trigger (e.g., "sued", "suing", "filed a
                lawsuit", "suit").
            ● plaintiff: The party bringing the suit. Strip out any articles or
                adjectives.
            ● defendant: The party being sued. Again, keep references minimal.
            ● adjudicator: The judge or court if one is explicitly named.
            ● crime: If a wrongdoing is stated (e.g., "for fraud", "for breach
                of contract").
            ● time: When the suit is filed or mentioned.
            ● place: Where the suit is taking place.

        What to do:
            - Label the party initiating the lawsuit as "plaintiff."

        What not to do:
            - Do not confuse "plaintiff" with "defendant" if the text clearly
                states who is suing whom.

        Example:
            Text: "A nurse sued Dell for bait and switch."
            → Sue(mention='sued', plaintiff=['nurse'], defendant=['Dell'],
                crime=['bait and switch'], time=[], place=[])
        """
        mention: str  # minimal expression for the lawsuit event
        plaintiff: List[str]  # who brings the suit
        defendant: List[str]  # who is being sued
        adjudicator: List[str]  # the judge or court, if stated
        crime: List[str]  # the wrongdoing for which the suit is filed
        time: List[str]  # when the suit took place
        place: List[str]  # where the suit took place

class Appeal(JusticeEvent):
        """
        An Appeal Event occurs whenever a court decision is taken to a higher
            court
        for review.

        Refined Guidelines:
            ● mention: The short trigger for the appeal (e.g., "appeal", "
                appealed").
```

```
        • defendant: The party accused or found guilty, if the text states
            so.
        • prosecutor: The party bringing the appeal (i.e., the appellant).
            This might be the same individual who was a defendant in a
            prior trial but is now appealing.
        • adjudicator: The higher court or judge handling the appeal, if
            given.
        • crime: The wrongdoing for which the appeal is made (if stated).
        • time: When the appeal is filed or heard.
        • place: Where the appeal is taking place.

    What to do:
      - If text says someone "filed an appeal," that entity is the "
          prosecutor" if no other roles are specified.
      - If the text does not identify an accused, keep defendant=[].

    What not to do:
      - Do not automatically fill "defendant" if it's unclear who was
          accused.

    Example:
      Text: "He appealed the verdict last week."
      → Appeal(mention='appealed', defendant=[], prosecutor=['He'], crime
          =[], time=['last week'], place=[])
    """
    mention: str  # minimal word for the appeal event
    defendant: List[str]  # the accused, if stated
    prosecutor: List[str]  # who is bringing the appeal
    adjudicator: List[str]  # the judge or court for the appeal
    crime: List[str]  # the crime or issue being appealed
    time: List[str]  # when the appeal occurs
    place: List[str]  # where the appeal is heard
```

## C.3 EXAMPLE OF TASK INSTRUCTION AND OPTIMAL EVENT GUIDELINES GENERATED BY GPT-4.5

```
# This is an event extraction task for identifying and structuring events
    from text using Python-defined event classes. Each structured event
    consists of an event trigger word, an event type, participant
    arguments, and their roles. Your objective is to output this
    information in a Python list of events, ensuring it is Python-
    parsable and strictly follows the event definitions provided below.

## Instructions:

1. **Span Extraction**:
    - Extract precise and concise spans for mentions and participant
        arguments, conveying the event or argument role clearly without
        unnecessary context.
    - For extracts involving titles or specifics, use general terms
        unless details are crucial to the events integrity.
    - When identifying entity roles in events, prioritize the core
        identifiers over accompanying descriptors.

2. **Role Identification**:
    - Accurately identify roles using contextual cues, effectively
        resolving ambiguities while prioritizing explicit spans. If roles
         are unmentioned, leave them empty.
    - Maintain consistency, particularly with distinctions like plaintiff
         vs. defendant, based on contextual evidence.
    - Clarify roles in complex transactions, such as distinguishing
        between beneficiaries and direct recipients.
```

```
3. **Output Format**:
    - Please follow the Python-format EventName("mention" = "trigger", "
        role1" = [...], "role2" = [...], ...) strictly.
    - Ensure consistent output in the specified format for Python
        compatibility, adhering strictly to event definitions.
    - Represent unmentioned participants with an empty list rather than
        assumptions or placeholders.

4. **Clarifications and Exceptions**:
    - Note explicitly when roles have exceptions based on role
        definitions.
    - Manage overlapping roles by following specific guidelines for span
        clarity and precision, ensuring no crucial details are overlooked
        .

5. **Consistency**:
    - Ensure consistency in role identification and event extraction
        across similar scenarios.
    - Address ambiguity and overlap by defining roles explicitly and
        setting clear precedence for extraction guidelines.

Below are the structured event definitions:

# Here are the event definitions:

class Convict(JusticeEvent):
    """
    A Convict Event signifies the successful prosecution of a defendant.
        This involves a person, organization, or geographical political
        entity (GPE) being convicted for a crime.
    """
    mention: str  # Focus on concise triggers like "convicted" or "
        conviction", avoiding embellishments.
    defendant: List[str]  # Name the convicted individuals or entities.
        Use direct identifiers, example: "John Doe".
    adjudicator: List[str]  # Reference the judicial entity, example: "
        court" or "judge", unless specifics are critical.
    crime: List[str]  # Provide short, precise descriptions of crimes, e.
        g., "fraud".
    time: List[str]  # Specify exact times if mentioned, e.g., "Monday".
    place: List[str]  # Note locations if explicitly mentioned, avoid
        assumptions.

class TransferMoney(TransactionEvent):
    """
    Non-purchasing money transfers involving giver and recipient roles,
        where transactions are more indirect or complex.
    """
    mention: str  # Use explicit terms like "donated", staying concise.
    giver: List[str]  # Identify the money source, example: "Sheila C.
        Johnson".
    recipient: List[str]  # Clearly name receiving entities.
    beneficiary: List[str]  # Note additional beneficiaries unambiguously
        .
    money: List[str]  # Use exact figures, avoiding vague amounts.
    time: List[str]  # Define occurrence times if clearly specified.
    place: List[str]  # Mention the transaction location if detailed.

class Meet(ContactEvent):
    """
    Events where entities gather face-to-face, e.g., meetings, summits,
        or conferences.
```

```
      """
    mention: str  # Central meeting references like "summit", without
        extra detail.
    entity: List[str] # List participants clearly, omitting superfluous
        descriptions.
    time: List[str]  # Specify times if explicitly provided.
    place: List[str]  # Mention locations if available, avoiding
        unsupported assumptions.

class PhoneWrite(ContactEvent):
    """
    Non-face-to-face communications, covering written and phone-based
        interactions.
    """
    mention: str  # Terms indicating communication, e.g., "called",
        succinctly.
    entity: List[str]  # Capture the participants in the communication.
    time: List[str]  # Specify times if mentioned, ensuring clarity.

class DeclareBankruptcy(BusinessEvent):
    """
    Occurs when an organization requests legal protection from debt
        collection.
    """
    mention: str  # Use declarations like "bankruptcy", clearly.
    org: List[str]  # Focus on the organizational name in question.
    time: List[str]  # Mention when the declaration occurs if explicitly
        stated.
    place: List[str]  # Note the declaration's location if outlined.

class EndOrg(BusinessEvent):
    """
    An organization ceases operations, going out of business completely.
    """
    mention: str  # Use terms like "shut down" to capture essence
        effectively.
    org: List[str]  # Succinctly list the organizations ending operations
        .
    time: List[str]  # Clearly mention when specifics are supplied.
    place: List[str]  # Mention location details if clearly stated.

class Die(LifeEvent):
    """
    Event marking the end of life, covering direct, accidental, and self-
        inflicted cases.
    """
    mention: str  # Specific terms like "died", excluding excess context.
    agent: List[str]  # Cite any responsible party if indicated.
    victim: List[str]  # Precisely identify the deceased without titles.
    instrument: List[str]  # Specify instruments used if described.
    time: List[str]  # Use accurate timing where provided.
    place: List[str]  # Mention locations where explicitly noted.

class SentenceAct(JusticeEvent):
    """
    Legal sentence issuance, often involving incarceration.
    """
    mention: str  # Direct words like "sentenced", retaining clarity.
    defendant: List[str]  # Identify the sentenced party succinctly.
    adjudicator: List[str]  # State the authority issuing the sentence.
    crime: List[str]  # Precisely include mentioned crimes.
    sentence: List[str]  # Clearly outline the penalties involved.
    time: List[str]  # Specific timing if explicitly declared.
    place: List[str]  # Cite location details when supplied.
```

```
1782   class Sue(JusticeEvent):
1783       """
1784       The initiation of legal proceedings against an entity to determine
1785          liability.
1786       """
1787       mention: str  # Specific terms like "sued".
           plaintiff: List[str]  # Clearly identify the suing parties.
1788       defendant: List[str]  # Identify the sued entities unambiguously.
1789       adjudicator: List[str]  # Specify judicial role if expressed.
1790       crime: List[str]  # Highlight alleged crimes if specified.
           time: List[str]  # Reference explicit timing if detailed.
1791       place: List[str]  # Extract the location details if outlined.
1792
1793   class Appeal(JusticeEvent):
1794       """
1795       Represents decisions moved to higher courts for further review.
1796       """
1797       mention: str  # Use terms like "appealed" directly.
1798       defendant: List[str]  # Name the entity under review.
           prosecutor: List[str]  # Name the initiating party of the appeal.
1799       adjudicator: List[str]  # Reference the reviewing court.
1800       crime: List[str]  # Clearly detail crimes if mentioned.
1801       time: List[str]  # Capture filing times if explicit.
           place: List[str]  # Mentioned locale of appeal if detailed.
1802
```

## C.4 EXAMPLE OF OPTIMAL TASK INSTRUCTION AND EVENT GUIDELINES GENERATED BY GPT-4O

```
# This is an event extraction task where the goal is to extract
    structured events from the text following structured event
    definitions in Python. A structured event contains an event trigger
    word, an event type, the arguments participating in the event, and
    their roles in the event.

# Task Instructions:
1. For each different event type, output the extracted information from
    the text into a Python list format where:
    - The first key 'mention' holds the value of the event trigger.
    - Subsequent keys/values follow the class definitions below.

2. Structure the output in a valid Pydantic format: `result = [EventName(
    "mention" = "trigger", "arg1_key" = "arg1_span", ...)]`.
3. Adhere strictly to the described event descriptions and role
    definitions, considering implicit contexts and indirect attributions.
4. Address special cases:
    - Appeals: Consider involved parties from prior related events as ``
       prosecutor''.
    - Multiple roles may apply contextually; ensure complete information
       extraction.
    - Implicit indications: If mentions like "filed", "concluded", etc.,
       suggest indirect roles, use context to clarify them.

5. Maintain backward compatibility where applicable. Do not output
    anything else except parsable structured event format in Python.

# Here are the event definitions:

class Convict(JusticeEvent):
    """
```

```
    A Convict Event occurs whenever a Try Event ends with a successful
        prosecution of the Defendant.
    There may not always be explicit mentions of crimes in the text; use
        contextual clues.
    """
    mention: str  # The text span that expresses the conviction (e.g., "
        convicted").
    defendant: List[str]  # The entity found guilty, search for adjacent
        terms like "defendant".
    adjudicator: List[str]  # The judge or court, often implicitly
        understood from context.
    crime: List[str]  # Crime references, even implied (e.g., "guilty of
        ...").
    time: List[str]  # When conviction happens, contextual or explicit
        dates.
    place: List[str]  # Where the conviction occurs, often a court or
        city name nearby.

class TransferMoney(TransactionEvent):
    """
    Refers to money transfer actions outside purchasing contexts.
        Recognize givers and recipients even in indirect mentions.
    """
    mention: str  # Turn of phrase indicating transfer (e.g., "
        transferred", "donated").
    giver: List[str]  # Entity initiating transfer (may be implied; use
        context).
    recipient: List[str]  # Direct receiver of money, often clearly
        stated.
    beneficiary: List[str]  # Can be implied; beneficiaries are often
        indirect.
    money: List[str]  # Described amounts; look for currency signs ($, €,
         etc.).
    time: List[str]  # Dates or relative times (e.g., "two years ago").
    place: List[str]  # Locations of transaction, if specified.

class Meet(ContactEvent):
    """
    Occurs when entities meet face-to-face; discern collective entity
        mentions from individual roles.
    """
    mention: str  # Trigger phrases (e.g., "meet", "conference").
    entity: List[str]  # Entities, clarified through context or explicit
        mentions.
    time: List[str]  # When entities meet, even if future planned.
    place: List[str]  # Meeting location, from nearby phrases.

class PhoneWrite(ContactEvent):
    """
    Encompasses non-face-to-face communications; cover implied
        interactors.
    """
    mention: str  # Non-direct communication identified triggers (e.g., "
        called", "emailed").
    entity: List[str]  # Communicating entities, occasionally understood
        indirectly.
    time: List[str]  # Times derived from text, even if not very specific
        .

class DeclareBankruptcy(BusinessEvent):
    """
    An event signifying financial distress declarations; distinguish from
         emergence narratives.
    """
    mention: str  # Indicators like "declared bankruptcy".
```

```
      org: List[str]  # Company/entity that declared, directly mentioned.
      time: List[str]  # Declaration date, often provided.
      place: List[str]  # Geographical context of declaration.

class EndOrg(BusinessEvent):
      """
      Concludes an organization's operations; ensure specificity of
          organization ceases.
      """
      mention: str  # Marks of closure (e.g., "dissolved", "shutdown").
      org: List[str]  # Organization ending, referenced in texts.
      time: List[str]  # Date context around organization ending.
      place: List[str]  # Location tied to organizational operations.

class Die(LifeEvent):
      """
      Recognizes cessation of life events; determine involvements from
          surrounding text.
      """
      mention: str  # Triggering term showing death (e.g., "died", "passed
          away").
      agent: List[str]  # Agents causing death if deliberate; contextual
          deductions.
      victim: List[str]  # Deceased, named or implied victims.
      instrument: List[str]  # Weapons or causes if mentioned.
      time: List[str]  # Death-related timing, even metaphorical.
      place: List[str]  # Place the death occurred, discerned from text.

class SentenceAct(JusticeEvent):
      """
      Legal actions culminating in punishment; include implied authority
          adjudication references.
      """
      mention: str  # Verbs indicating sentencing (e.g., "sentenced").
      defendant: List[str]  # Persons sentenced, more direct mentions.
      adjudicator: List[str]  # State actor issuing punishment.
      crime: List[str]  # Crimes specified can be explicit or by context
          related.
      sentence: List[str]  # Detailed punishments, commonly listed.
      time: List[str]  # Contextual timing of legal processes.
      place: List[str]  # Legal venues, stated or implicit.

class Sue(JusticeEvent):
      """
      Legal actions initiation detections; interpreting mentions to detect
          implicated parties.
      """
      mention: str  # Lawsuit trigger terms (e.g., "sued").
      plaintiff: List[str]  # Agents initiating, even implicit from context
          .
      defendant: List[str]  # Specific subjects of the lawsuit.
      adjudicator: List[str]  # Legal bodies, typically explicit.
      crime: List[str]  # Charges or offenses underpinning the suit.
      time: List[str]  # Suit filing and related timings.
      place: List[str]  # Locations cited, often courts.

class Appeal(JusticeEvent):
      """
      Reviewal legal challenges; correctly attribute events around
          appellate actions.
      """
      mention: str  # Terms denoting appeals like "appealed".
      defendant: List[str]  # Party whose case goes under review.
      prosecutor: List[str]  # Original case actors initiating the appeal,
          inferred.
```

```
adjudicator: List[str]  # Higher court taking the over evaluation.
crime: List[str]  # Reviews' subject offenses.
time: List[str]  # Appeal reference times, may not be given.
place: List[str]  # Court location details or broader judicial zones.
```

