# OpenReview forum: "Revisiting Prompt Optimization with Large Reasoning Models---A Case Study on Event Extraction"
_ICLR.cc/2026/Conference — Submitted to ICLR 2026_

### Official Review · Reviewer_y32Z · 2025-10-16

**Soundness:** 2
**Presentation:** 2
**Contribution:** 1
**Rating:** 2
**Confidence:** 5

**Summary:**

* **Research Problem**:
  Investigates whether Large Reasoning Models (LRMs), despite their strong inherent reasoning capabilities, still benefit from prompt optimization, and whether they can serve as effective prompt optimizers—using event extraction as a structured testbed.

* **Experimental Method**:
  Evaluates two LRMs (DeepSeek-R1, OpenAI o1) and two LLMs (GPT-4.5, GPT-4o) in both task-model and prompt-optimizer roles within a Monte Carlo Tree Search (MCTS) framework; experiments are conducted on event extraction (ACE05), symbolic reasoning, and biomedical NER tasks.

* **Main Findings**:

  1. LRMs significantly benefit from prompt optimization, with greater performance gains than LLMs.
  2. LRMs (especially DeepSeek-R1) outperform LLMs as prompt optimizers, producing higher-quality, more stable, and more concise prompts.
  3. These benefits generalize beyond event extraction to other domains, including symbolic and biomedical tasks.

**Strengths:**

1. The paper revisits prompt optimization in the context of Large Reasoning Models (LRMs) which is a meanful topic. It challenges the prevailing assumption that strong reasoning models no longer require prompt optimization, offering a novel empirical perspective.


2. The main insight of the experiment are constructive: the performance of LRM can be further improved by optimizing prompts.

**Weaknesses:**

1. Although the authors have conducted meaningful empirical research, this paper does not make any original theoretical or experimental contributions.

2. The overall experimental focus of this paper remains primarily on event extraction. Evaluation on a wider range of NLP tasks would enhance the significance of the paper's conclusions.

3. The conclusions of this paper are highly dependent on the test model and test task. LLMs are highly dependent on training data, and LRMs are primarily optimized for complex reasoning tasks. Do similar conclusions hold for the recent SOTA models o3 and o4? For datasets that are already saturated, optimizing prompt words will obviously not bring any improvement. For extremely difficult problems, such FrontierMath or ARC-AGI, can optimizing prompt also bring improvement?

**Questions:**

1. Can the authors generalize the evaluation task to a wider range of NLP tasks, not just event extraction?
2. Do similar conclusions hold for the recent SOTA LRMs?
3.  Do similar conclusions hold for extremely difficult problems, such FrontierMath or ARC-AGI?

---

> ### Author Response · Authors · 2025-11-25
> **Response to Reviewer y32Z**
>
> We thank the reviewer for carefully reading our work and for recognizing the importance of revisiting prompt optimization, specifically in the context of Large Reasoning Models (LRMs). We appreciate the reviewer highlighting that (i) this is a meaningful and timely problem, (ii) our empirical findings challenge the popular assumption that LRMs no longer require prompt engineering, and (iii) the experimental insights are constructive in showing that LRMs can further improve through optimized prompts. We address the reviewer’s concerns regarding contribution, task coverage, and generality across model families and difficulty regimes below.
>
> ## W1: No original theoretical or experimental contributions.
>
> We respectfully disagree. As clarified in General Response G1, the novelty of this work is conceptual and empirical rather than algorithmic. Prior prompt-optimization research has exclusively examined LLMs, whereas our paper provides the first systematic study of how LRMs behave under discrete prompt optimization, a setting that did not exist in the literature before. This yields several findings that are new, non-obvious, and practically important, including: (i) LRMs still benefit substantially from optimization; (ii) LRMs act as stronger prompt optimizers than LLMs; (iii) LRMs converge earlier and more stably in MCTS; and (iv) LRMs surface latent rule structures and show disproportionately large gains in low-resource regimes.
>
> These results constitute new empirical knowledge about reasoning-capable models, and represent the primary contribution of the paper. Additional discussion of conceptual novelty is provided in General Response G1.
>
> ## W2: The paper focuses primarily on event extraction; it needs wider NLP tasks.
>
> We want to clarify that our experiments already extend well beyond event extraction. As discussed in Lines 073–076 of the introduction and in Lines 406–421 (RQ5), we explicitly evaluate on:
>
> (1) Geometric Shapes: a symbolic reasoning task requiring multi-step spatial deduction, structurally different from EE and not schema-driven;
>
> (2) NCBI Disease NER: a biomedical entity-recognition task with natural-language prompts and no event schemas; and
>
> (3) (Added in rebuttal G2) the Agzhal et al. [1] 6×6 planning benchmark, which is entirely outside the NLP/EE domain and tests open-ended multi-step reasoning.
>
> Across all three non-EE settings, we observe the same qualitative behavior: LRMs benefit more from optimization, converge more smoothly, and act as stronger prompt optimizers. The additional planning experiment further reinforces that our conclusions are not tied to any linguistic property of EE, but reflect broader LRM optimization dynamics.
>
> A detailed summary of cross-task generalization is provided in General Response G2.

---

> ### Author Response · Authors · 2025-11-25
> **Response to Reviewer y32Z (continued)**
>
> ## W3: Concern about dependence on specific models/tasks and questions about o3/o4, saturated datasets, and frontier benchmarks.
>
> We address each part of this concern in turn.
>
> **(1) Are the conclusions tied to a particular model or task?**
>
> As discussed in the main paper and General Response G1-G2, our findings are consistent across:
> - Four models in the original submission (GPT-4o, GPT-4.5, DeepSeek-R1, o1),
>
> - Four distinct domains: structured EE (ACE05), symbolic reasoning (Geometric Shapes), biomedical NER (NCBI Disease), and open-ended planning [1].
>
> In all of these settings, we observe the same qualitative pattern: (1) LRMs benefit substantially from prompt optimization, (2) LRMs act as stronger optimizers than LLMs, and (3) Optimized prompts are shorter, more stable, and more rule-grounded.
>
> This cross-model, cross-task consistency suggests that our conclusions reflect class-level behavior of LRMs, rather than an artifact of a single dataset or architecture.
>
> **(2) Do similar conclusions hold for newer LRMs such as o3/o4?**
>
> To directly address this, we additionally evaluated OpenAI **o3** and **o4** on ACE_med using the same MCTS-based optimization framework as in the main experiments. The results closely mirror the behavior of DeepSeek-R1 and o1:
>
> | Model | No Opt. (test) | Depth-1 (dev) | Depth-5 (dev) | Depth-5 (test) |
> |-------|----------------|---------------|---------------|----------------|
> | o3    | 19.63          | 25.61         | 48.95         | 45.00          |
> | o4    | 20.25          | 28.53         | 50.11         | 46.51          |
>
> In both cases, we observe substantial and monotonic improvements with increased optimization depth (e.g., No Opt -> Depth-5 on test: +25.37 for o3 and +26.26 for o4), and Depth-5 significantly outperforms Depth-1 on the dev set. This pattern is fully consistent with our findings for DeepSeek-R1 and o1, and confirms that **newer SOTA LRMs also benefit meaningfully from prompt optimization and continue to act as strong prompt optimizers**. We will include these extended results in the final version.
>
> **(3) Addressing comment: “For datasets that are already saturated, optimizing prompts will obviously not bring improvement.”**
>
> We fully agree with the reviewer that on truly saturated datasets, where models already perform at or near 100%, there is little room for any method (including prompt optimization) to improve performance. However, our chosen benchmarks are deliberately not in this regime.
> For example, on ACE_med with DeepSeek-R1 as the task model:
> (1) No Opt AC on the test set is only 16.0,
> (2) Optimization raises AC to 35.2 at depth-1 and 43.75 at depth-5.
>
> This clearly shows that, despite ACE05 being an older dataset, it is far from saturated for current LRMs/LLMs in our schema-grounded, prompt-based EE formulation. The same holds for Geometric Shapes, NCBI, and the planning task, where zero-shot performance is well below ceiling and optimization yields large gains.
>
> Even if some training data for these models overlaps with the underlying corpora, the substantial deltas we observe indicate that the specific structured prompting + optimization setup is not saturated and that our findings are meaningful in this regime.
>
> **(4) What about extremely difficult benchmarks like FrontierMath or ARC-AGI?**
>
> We agree that frontier-level reasoning benchmarks (e.g., FrontierMath, ARC-AGI) are important future directions. However, they require specialized pipelines that fall
> outside the scope of a unified, text-based MCTS optimization framework:
> - FrontierMath is not fully public: the complete dataset and official evaluation infrastructure are unavailable, and ground-truth solutions are not released, making it incompatible with reward-based prompt optimization.
> - ARC-AGI is non-textual and requires program-synthesis-style solvers, vision-to-symbol translation, and DSL induction, components that cannot be expressed through the textual prompt interface used in our study.
>
> Our goal in this paper is to characterize how LRMs interact with prompt optimization across a diverse but tractable set of tasks (structured EE, symbolic reasoning, biomedical IE, and planning). We will explicitly mention frontier-level benchmarks as promising future extensions, but note that this is orthogonal to our central claim: that LRMs, even with strong inherent reasoning capabilities, still benefit substantially from input-side optimization and act as strong prompt optimizers across multiple domains.

---

> ### Author Response · Authors · 2025-11-25
> **Response to Reviewer y32Z (continued)**
>
> ## Q1. Can the authors generalize the evaluation task to a wider range of NLP tasks, not just event extraction?
>
> Please refer to General Response G2.
>
> ## Q2. Do similar conclusions hold for the recent SOTA LRMs?
>
> Please refer to W3.
>
> ## Q3. Do similar conclusions hold for extremely difficult problems, such FrontierMath or ARC-AGI?
>
> Please refer to W3.
>
> **References**
>
> [1] Mohamed Aghzal, Erion Plaku, & Ziyu Yao (2024). Can Large Language Models be Good Path Planners? A Benchmark and Investigation on Spatial-temporal Reasoning. In ICLR 2024 Workshop on Large Language Model (LLM) Agents.

---

### Official Review · Reviewer_cQqu · 2025-10-29

**Soundness:** 2
**Presentation:** 3
**Contribution:** 2
**Rating:** 4
**Confidence:** 5

**Summary:**

This paper presents a timely and systematic empirical study investigating whether the advanced reasoning capabilities of Large Reasoning Models (LRMs) like DeepSeek-R1 and o1 diminish the need for prompt optimization, using the complex task of event extraction as a primary case study; the authors claim that LRMs still benefit significantly from optimization, that they serve as more effective prompt optimizers than general-purpose LLMs, and that these findings generalize to other tasks like symbolic reasoning and biomedical NER.

**Strengths:**

(1) Clear motivation and timely question: The paper addresses a relevant and open question in the era of advanced reasoning models: whether prompt engineering remains necessary. This is especially valuable as the community increasingly adopts LRMs without fully understanding their interaction with prompting strategies.

(2) Rigorous experimental design: The use of a unified MCTS-based prompt optimization framework allows fair comparison across models as both task solvers and optimizers. The inclusion of low- and medium-resource settings, depth-controlled MCTS rollouts, and cross-task generalization strengthens the empirical foundation.

(3) Comprehensive analysis: The paper includes convergence curves, survival plots, error categorization, and qualitative prompt comparisons, offering multiple lenses to interpret results. The observation that DeepSeek-R1 achieves high performance with shorter prompts is insightful.

**Weaknesses:**

(1) Limited Methodological Novelty: The core optimization algorithm (MCTS) is adopted from prior work (e.g., PromptAgent). The primary novelty lies in its application to LRMs rather than in a fundamental advancement of the optimization technique itself. The paper is more of a thorough empirical benchmark than a methodological contribution.

(2) Task selection bias: Event extraction is a highly structured, schema-constrained task. While the authors test generalization on two other tasks, the main conclusions are anchored in a setting where explicit guidelines and code-based prompting play an outsized role. It remains unclear whether the observed LRM advantages would hold in more open-ended reasoning tasks (e.g., mathematical proof, planning).

(3) Incremental Nature of Key Finding: The central finding that more capable models benefit from optimized prompts is intuitively plausible and, to some extent, expected. While the quantitative demonstration is valuable, it may not be sufficiently surprising or groundbreaking for a top-tier venue.

**Questions:**

(1) Beyond the application of an existing MCTS framework to a new model class (LRMs), what is the core conceptual or methodological novelty of this work that distinguishes it from prior prompt optimization research?

(2) How confident are you that the observed advantages of LRMs would hold on the full ACE05 dataset with all 33 event types? Did you run any preliminary experiments that suggested context length would become a major bottleneck?

(3) In the Geometric Shapes and NCBI tasks, did you use the same code-based prompting format? If not, how was the prompt structure adapted, and could that influence the observed generalization?

---

> ### Author Response · Authors · 2025-11-25
> **Response to Reviewer cQqu**
>
> We thank the reviewer for the thoughtful and detailed evaluation and for highlighting several strengths of our work. We appreciate the reviewer recognizing (i) the clear motivation behind studying prompt optimization in the era of LRMs, (ii) the rigor and fairness of our unified MCTS-based experimental design, and (iii) the comprehensive analyses, such as convergence curves, survival plots, and qualitative prompt comparisons. Below, we address the reviewer’s concerns regarding novelty, task selection, and generality.
>
> ## W1: Limited Methodological Novelty
>
> Please refer to general response G1 for new results and conclusions.
>
>
> ## W2: Concern about task selection bias and whether results hold in open-ended reasoning (e.g., planning).
>
> Please refer to general response G2 for new results and conclusions.
>
>
> ## W3: Incremental Nature of Key Finding
>
> We appreciate the reviewer raising this point. While it may seem intuitive in hindsight that LRMs could still benefit from optimized prompts, our study shows that this assumption is neither trivial nor previously demonstrated. In fact, the prevailing belief in the community, including documentation for modern reasoning models and prior work such as [1, 2, 3], has been that LRMs largely eliminate the need for prompt engineering. Our results overturn this assumption across three distinct domains (structured EE, symbolic reasoning, biomedical NER) and, additionally, in open-ended planning [4], where we observe +30 to +34 point gains from prompt optimization **(please see General Response G2)**.
>
> Moreover, our findings reveal non-obvious behavioral patterns that were not predicted by prior literature: (1) LRMs converge earlier and more smoothly in MCTS; (2) they produce shorter but higher-quality optimized prompts; (3) they benefit disproportionately in low-resource regimes; and (4) they surface latent domain rules not present in human guidelines. None of these phenomena follow from the intuition that “better models benefit from prompting,” and they provide new insight into how reasoning models internalize and refine domain structure.
>
> Taken together, the contribution of this work is not simply that “optimization helps better models,” but that the interaction between optimization depth, resource regime, model class, and prompt complexity behaves fundamentally differently for LRMs than for LLMs, a previously unexplored regime that we characterize systematically. We will clarify this framing in the final version.
>
>
> ## Q1:  Beyond applying an existing MCTS framework to LRMs, what is the core conceptual or methodological novelty of this work?
>
> Please refer to G1, G2, and W3 for full details. In summary, our novelty does not lie in proposing a new search algorithm, but in providing the first systematic study of prompt optimization in the era of LRMs, a setting not examined in prior work. These results collectively provide new conceptual and behavioral insights into how reasoning-capable models interact with input-side optimization.
>
>
> ## Q2: How confident are you that the observed LRM advantages would hold on the full 33-type ACE05 schema? Did you run preliminary experiments suggesting that context length becomes a bottleneck?
>
> Please refer to W2 and General Response G3 for additional context. In brief, we did perform feasibility checks while constructing full-schema prompts (e.g., assembling the 33-type schema, measuring token length, and running dry MCTS rollouts). These checks revealed that full-schema prompting introduces long-context degradation effects that are orthogonal to the reasoning behavior we aim to evaluate.
>
> As discussed in Lines 212–215, including all 33 ACE05 event types produces extremely long prompts with multiple structured regions (schema, rules, examples, optimization feedback). Prior work on long-context processing [6, 7] shows that such inputs often experience attention dilution and retrieval failures even when the prompt fits within nominal context limits. In that regime, the primary bottleneck becomes prompt length rather than reasoning ability, which is not the focus of our study.
>
> Following standard schema-based EE practice [8], we therefore use reduced subsets (ACE_low, ACE_med) to avoid these confounders. Importantly, all four models evaluated (GPT-4o/4.5, DeepSeek-R1, o1) operated far below their context windows in our 10-type setting, and all showed consistent gains. This indicates that the observed advantages of LRMs arise from their optimization behavior, not prompt conciseness or context size. Future work will explore full-schema optimization as longer-context LRMs become more widely available.

---

> > ### Comment · Reviewer_cQqu · 2025-11-28
> >
> > Thank you for the response. While I acknowledge the authors’ effort in applying existing MCTS-based prompt optimization frameworks, originally developed for LLMs to LRMs, I remain unconvinced that this constitutes a contribution sufficient for ICLR. Therefore, I maintain my original rating.

---

> ### Author Response · Authors · 2025-11-25
> **Response to Reviewer cQqu (continued)**
>
> ## Q3: In the Geometric Shapes and NCBI tasks, did you use the same code-based prompting format? If not, how was the prompt structure adapted, and could that influence generalization?
>
> For Geometric Shapes and NCBI Disease NER, we used the original natural-language prompts provided by the dataset authors, rather than the code-style schema used in ACE’05. We intentionally retained their native prompt format to test whether our framework generalizes across both structured (schema-driven) and unstructured (natural-language) prompting styles.
>
> For ACE’05 specifically, we followed established prior approaches in event extraction that incorporate the schema definitions and annotation guidelines directly into the prompt. This code-style representation is standard in recent generative EE work and is necessary for conveying the detailed event-type rules that natural-language prompts typically omit.
>
> Despite these differences in surface form, the qualitative trends remained identical across all three domains: LRMs benefited more from prompt optimization, converged more smoothly, and produced higher-quality optimized prompts than LLMs. This is consistent with observations from PromptAgent and other optimization studies that report stable improvements across heterogeneous prompt formats.
>
> Thus, while the prompt styles naturally differ across tasks due to domain conventions, the core finding “LRMs benefiting more and acting as stronger optimizers” remains stable, indicating that our findings do not depend on ACE-style code-based prompting.
>
> **References**
>
> [1] Wang, Guoqing, et al. "Do advanced language models eliminate the need for prompt engineering in software engineering?." ACM Transactions on Software Engineering and Methodology (2024).
>
> [2] OpenAI. (2025). Reasoning Best Practices. Url: https://platform.openai.com/docs/guides/reasoning-best-practices#how-to-prompt-reasoning-models-effectively
>
> [3] Agustin Mantaras. (2025). Prompt Engineering for OpenAI’s O1 and O3-mini Reasoning Models. URL: https://techcommunity.microsoft.com/blog/azure-ai-services-blog/prompt-engineering-for-openai%E2%80%99s-o1-and-o3-mini-reasoning-models/4374010
>
> [4] Wenxuan Liu, Zixuan Li, Long Bai, Yuxin Zuo, Daozhu Xu, Xiaolong Jin, Jiafeng Guo, and Xueqi Cheng. 2025. Towards Event Extraction with Massive Types: LLM-based Collaborative Annotation and Partitioning Extraction. In Proceedings of the 2025 Conference on Empirical Methods in Natural Language Processing, pages 34365–34387, Suzhou, China. Association for Computational Linguistics.
>
> [5] Mohamed Aghzal, Erion Plaku, & Ziyu Yao (2024). Can Large Language Models be Good Path Planners? A Benchmark and Investigation on Spatial-temporal Reasoning. In ICLR 2024 Workshop on Large Language Model (LLM) Agents.
>
> [6] Zhang, H., Feng, T., Han, P., & You, J. (2025). Academiceval: Live long-context llm benchmark.
>
> [7] Liu, Nelson F., et al. "Lost in the middle: How language models use long contexts." Transactions of the Association for Computational Linguistics 12 (2024): 157-173.
>
> [8] Saurabh Srivastava, Sweta Pati, and Ziyu Yao. 2025. Instruction-Tuning LLMs for Event Extraction with Annotation Guidelines. In Findings of the Association for Computational Linguistics: ACL 2025, pages 13055–13071, Vienna, Austria. Association for Computational Linguistics.

---

### Official Review · Reviewer_AcPa · 2025-10-31

**Soundness:** 3
**Presentation:** 3
**Contribution:** 2
**Rating:** 6
**Confidence:** 3

**Summary:**

The paper studies whether Large Reasoning Models (LRMs; DeepSeek-R1, OpenAI o1) still benefit from prompt optimization and whether they are good optimizers themselves, using event extraction (ACE05) as a case study within an MCTS framework. The main finding is that optimization helps all models but LRMs benefit most, both as task models and as optimizers. The claim is probed further with convergence/quality analyses and two non-EE tasks.

**Strengths:**

S1 - Clear problem framing and technically sound MCTS setup with explicit four-step loop.

S2 - 2. Strong, easy-to-grasp headline result that LRMs both benefit more from optimization and optimize better than LLMs.

S3 - 1. Sensible experimental design with two data regimes (ACElow/ACEmed) and two evaluation depths (depth 1 vs depth 5), enabling controlled comparison/

**Weaknesses:**

W1 - Metric mismatch in optimization v reporting -- The reward aggregates averaged F1 across TI/TC/AI/AC (s. 3.2), yet the analysis "primarily reports AC" (I understand the authors provide a citation for this choice but I find it unsatisfactory), creating a potential objective-reporting mismatch. Clarify why not optimize AC directly.

W2 - Downsampling schema may bias conclusions -- To avoid (presumably) long prompts, the paper downsamples ACE05 to 10 event types and leaves long-contextg processing to future work (L236-245). This choice may favor models preferring concise prompts and limits external validity to full-schema EE.

W3 - Depth-5 gains are modest, at best. Paper does note "non-dramatic" improvements from full-depth over d-1 (RQ2) and tab 1 shows small deltas. This sort of raises questions about the practical alue of deeper search.

W4 - No statistical uncertainty reported. Main tables/figures lack confidence intervals or sig tests, making it hard to judge robustness of improvements.

**Questions:**

Q1. Why optimize the average of TI/TC/AI/AC instead of AC directly, given AC is your primary metric? Any evidence that the averaged reward improves AC more than AC-only reward?

Q2. How were the 10 ACE05 event types chosen, and do conclusions hold on the full 33-type schema (or with long-context methods)?

---

> ### Author Response · Authors · 2025-11-25
> **Response to Reviewer AcPa**
>
> We thank the reviewer for the clear summary and for recognizing the strengths of our setup, particularly in problem framing, the explicit MCTS formulation, and the controlled comparisons across data regimes and search depths. We also appreciate the reviewer highlighting the central result that LRMs both benefit more from optimization and act as stronger optimizers. Below, we address the reviewer’s specific concerns regarding metric design, schema subsampling, depth analysis, and robustness reporting.
>
> ## W1: Concern about reward metric mismatch (averaged F1 vs. reporting AC).
>
> We thank the reviewer for this helpful observation. Our choice of reward and reporting metrics is deliberate and grounded in the structure of event extraction (EE).
>
> **(1) Why do we optimize the average F1 (TI/TC/AI/AC)?**
>
> EE is an inherently pipeline-coupled task: Trigger Identification -> Trigger Classification -> Argument Identification -> Argument Classification.
> Errors in earlier subtasks propagate into AC; e.g., weaker TI or TC directly lowers AC because missing or misclassified triggers eliminate all downstream arguments.
> Optimizing only AC would therefore encourage the optimizer to overfit the final subtask while neglecting the foundational ones, creating degenerate prompts (e.g., prompts that classify arguments well only when triggers are perfect).
> Optimizing the averaged F1 yields balanced gradients across subtasks, stabilizes search, and empirically produces strictly stronger AC results.
> We verified this via ablations using DeepSeek-R1 as both optimizer and task model, at depth-1 and depth-5:
>
> **Depth-1 Ablation Table**
>
> | Reward Type              | TI    | TC    | AI    | AC    |
> |--------------------------|--------|--------|--------|--------|
> | **Average (ours)**       | 37.50 | 33.93 | 25.00 | 24.66 |
> | **AC-only reward**       | 33.26 | 31.09 | 18.78 | 17.37 |
> | **Weighted (.3/.3/.2/.2)** | 41.89 | 37.57 | 23.08 | 21.72 |
>
> **Depth-5 Ablation Table**
>
> | Reward Type              | TI    | TC    | AI    | AC    |
> |--------------------------|--------|--------|--------|--------|
> | **Average (ours)**       | 56.60 | 56.60 | 44.26 | 44.26 |
> | **AC-only reward**       | 44.44 | 44.44 | 30.11 | 29.03 |
> | **Weighted (.3/.3/.2/.2)** | 59.26 | 59.26 | 37.58 | 36.24 |
>
> (**Note:** the weighted reward is a diagnostic baseline to test whether giving slightly higher emphasis to upstream subtasks (TI/TC), which directly constrain AI and AC, would alleviate the collapse seen under AC-only optimization, but the weights are not fine-tuned.)
>
> We make the following observations across both depths:
>
> - AC-only optimization performs worst because the model deprioritizes TI/TC, causing a cascading drop in AI and AC.
>
> - Weighted variants perform better only when TI/TC receives a higher weight, reflecting the pipeline dependency.
>
> - Averaged reward yields the strongest and most stable overall results, confirming that equal weighting is the most principled choice for EE.
>
> These results confirm that equal weighting is not arbitrary; it is the only reward structure that preserves the dependencies inherent in EE.
>
> **(2) Why is AC the primary reported metric in the main text?**
>
> AC is the standard headline metric in prior EE work [1] and is the subtask most sensitive to reasoning quality, hence its prominence in the main paper.
> However, **we do report full TI/TC/AI/AC breakdowns for every model and depth in Appendix B (Lines 237–239), and results on Geometric Shapes (accuracy) and NCBI NER (micro-F1) are also fully included.**

---

> ### Author Response · Authors · 2025-11-25
> **Response to Reviewer AcPa (continued)**
>
> ## W2: Concern about downsampling to 10 event types and potential bias.
>
> We refer the reviewer to General Response G3 for discussion.
>
> ## W3: Concern that depth-5 gains are modest and may question the value of deeper search.
>
> We thank the reviewer for the careful observation. The primary aim of our study is to determine whether LRMs still benefit from prompt optimization at all, not to show that deeper MCTS search yields dramatic improvements. Across all models and tasks, depth-1 already provides a significant lift, and depth-5 consistently improves further. This confirms that the optimization phenomenon itself is beneficial for LRMs.
>
> **Why do depth-5 gains appear modest for AC?**
>
> Argument Classification (AC) is the final stage of a pipeline-coupled task where improvements are naturally bottlenecked by upstream components. Prior EE work (e.g., [1, 2]) also reports diminishing returns on AC even with much larger models, indicating that AC is a fundamentally constrained subtask in ACE05.
>
> **Depth-5 does yield substantial gains where deeper search actually matters**
>
> *As shown in Appendix D.1*, TI and TC, EE’s foundational subtasks, improve by ~30% and ~20% respectively when moving from depth-1 to depth-5. These improvements are meaningful because higher-quality TI/TC directly enable all downstream subtasks, and they provide more reliable supervision signals for prompt refinement.
>
> **Why not explore an even deeper search?**
>
> Our depth-5 configuration (5 levels, 12 iterations, 3 children per node) was chosen to balance effectiveness and the very high inference cost of large optimizers such as GPT-4.5 and o1. Prior work [3] shows that deeper searches (depth >= 6) can discover even stronger prompts across multiple datasets, but such depths become computationally prohibitive for our LRM-based optimizer setting. Thus, depth-5 should be viewed as a computationally feasible midpoint, not an upper bound.
>
> ## Key takeaway
>
> Depth-5 gains are modest only for AC, the hardest and most bottlenecked subtask, but they are substantial for TI and TC, and they confirm that LRMs continue to refine prompts when given additional search depth. Since our primary research question is whether LRMs benefit from (and perform) prompt optimization at all, the results support the central claim of the paper.
>
>
> ## W4: Lack of statistical uncertainty in the main tables.
>
> We thank the reviewer for highlighting the importance of reporting statistical uncertainty. To address this, we computed 95% bootstrap confidence intervals for our core configuration, DeepSeek-R1 on ACE_med, using the same dev/test partitions, predictions, and evaluation code as in the main experiments. We followed standard practice and computed non-parametric bootstrap confidence intervals [4] by resampling the test set with replacement and recomputing the evaluation metric using the stored predictions. For each setting (No Opt, Depth-1, Depth-5), we resampled the test set with replacement and recomputed AC on each sample.
>
> | Setting    | AC (Test) | 95% CI             | Δ vs No Opt | Δ vs Depth-1 |
> |------------|-----------|--------------------|-------------|--------------|
> | No Opt     | **16.00** | 13.50 – 17.60      | -           | -            |
> | Depth-1    | **35.20** | 32.53 – 39.62      | **+19.20**  | -            |
> | Depth-5    | **43.75** | 40.80 – 47.60      | **+27.75**  | **+8.55**     |
>
> **Interpretation** The confidence intervals are narrow, and the improvements from optimization are substantially larger than their widths. For example, the gain from No Opt -> Depth-1 (+19.20 points) and No Opt -> Depth-5 (+27.75 points) both far exceed the uncertainty ranges (< ~4 points). Even Depth-1 -> Depth-5 (+8.55 points) exceeds the combined CI widths, indicating a meaningful and robust improvement. These results confirm that the observed gains are not due to sampling noise, but reflect consistent model behavior on the ACE_med test set
>
> ## Clarification of position
> Our intention is not to make claims about statistical micro-effects of different search depths, but rather to show that LRMs consistently benefit from prompt optimization, and that these benefits are stable even under test-set resampling. We will include confidence intervals for all main configurations in the revised version to more explicitly convey robustness.

---

> ### Author Response · Authors · 2025-11-25
> **Response to Reviewer AcPa (continued)**
>
> ## Q1 Why optimize the average of TI/TC/AI/AC instead of AC directly, given AC is your primary metric? Any evidence that the averaged reward improves AC more than the AC-only reward?
>
> Please refer to W1 for a detailed discussion. In summary, optimizing AC alone encourages the optimizer to overfit the last subtask while degrading TI/TC, which directly suppresses AC due to error propagation. On the contrary, equal-weighted averaging provides balanced learning signals across subtasks and avoids degenerate prompts that only work under perfect triggers.
>
> ## Q2 How were the 10 ACE05 event types chosen, and do conclusions hold on the full 33-type schema (or with long-context methods)?
>
> We refer the reviewer to General Response G3 for discussion. In addition, to address other concerns raised by the reviewer, we address them here separately:
>
> - **Why not use all 33 event types?**
> As explained in General Response G3 and Lines 212–215, including all 33 types in a single prompt shifts the bottleneck from reasoning to long-context prompt processing. Prior work [6,7] shows that prompts containing many structured regions (schemas, guidelines, examples, feedback) suffer attention and retrieval degradation even when they fit within nominal context limits.
>
>   Our goal is to evaluate models’ reasoning under optimization, not their robustness to long-context fragmentation, so we intentionally avoid this confounder.
>
> - **Are results driven by context-length sensitivity?**
> No. All four models evaluated in our study (GPT-4o/4.5, DeepSeek-R1 at 128k, and o1 at 200k) operated far below their context budgets in the 10-type setting, and all showed consistent gains. This indicates that the observed LRM advantages arise from their *optimization behavior*, not from prompt conciseness or context size.
>
> - **Does the reduced schema limit generality?**
> We observe the same qualitative trends on two additional domains with very different structures: Geometric Shapes (symbolic reasoning) and NCBI Disease NER (biomedical IE). In both cases, (i) LRMs benefit more from optimization, and (ii) LRMs serve as stronger optimizers. This supports that our findings are not tied to the reduced ACE schema but reflect broader interactions between reasoning capacity and input-side optimization.
>
>
> **References**
>
> [1] Kuan-Hao Huang, I-Hung Hsu, Tanmay Parekh, Zhiyu Xie, Zixuan Zhang, Prem Natarajan, Kai-Wei Chang, Nanyun Peng, and Heng Ji. 2024. TextEE: Benchmark, Reevaluation, Reflections, and Future Challenges in Event Extraction. In Findings of the Association for Computational Linguistics: ACL 2024, pages 12804–12825, Bangkok, Thailand. Association for Computational Linguistics.
>
> [2] Wenxuan Liu, Zixuan Li, Long Bai, Yuxin Zuo, Daozhu Xu, Xiaolong Jin, Jiafeng Guo, and Xueqi Cheng. 2025. Towards Event Extraction with Massive Types: LLM-based Collaborative Annotation and Partitioning Extraction. In Proceedings of the 2025 Conference on Empirical Methods in Natural Language Processing, pages 34365–34387, Suzhou, China. Association for Computational Linguistics.
>
> [3] Xinyuan Wang, Chenxi Li, Zhen Wang, Fan Bai, Haotian Luo, Jiayou Zhang, Nebojsa Jojic, Eric Xing, & Zhiting Hu (2024). PromptAgent: Strategic Planning with Language Models Enables Expert-level Prompt Optimization. In The Twelfth International Conference on Learning Representations.
>
> [4] Efron, Bradley, and Robert J. Tibshirani. An introduction to the bootstrap. Chapman and Hall/CRC, 1994.
>
> [5] Saurabh Srivastava, Sweta Pati, and Ziyu Yao. 2025. Instruction-Tuning LLMs for Event Extraction with Annotation Guidelines. In Findings of the Association for Computational Linguistics: ACL 2025, pages 13055–13071, Vienna, Austria. Association for Computational Linguistics.
>
> [6] Zhang, H., Feng, T., Han, P., & You, J. (2025). Academiceval: Live long-context llm benchmark.
>
> [7] Liu, Nelson F., et al. "Lost in the middle: How language models use long contexts." Transactions of the Association for Computational Linguistics 12 (2024): 157-173.

---

### Official Review · Reviewer_2nbd · 2025-11-02

**Soundness:** 4
**Presentation:** 4
**Contribution:** 3
**Rating:** 8
**Confidence:** 3

**Summary:**

The paper studies the interaction of prompt optimization for Large Reasoning Models (LRMs). They identify that this is a gap in existing literature which has only studied prompt optimization for Large Language Models (LLMs). In this work, DeepSeek-R1 and o1 are representative LRMs while GPT-4.5 and GPT4o are representative LLMs.

In particular, the paper focuses on structured prediction tasks since performance on these tasks is not yet saturated even with LRMs. The core analysis is conducted on the ACE05 Event Extraction task with supplementary analysis on Geometric Shapes and NCBI Disease NER tasks to show the generality of the findings. The paper uses a MCTS-based discrete prompt optimization algorithm with different LLMs/LRMs plugged in as the optimizer.

The core finding is that LRMs benefit from prompt optimization in both low and medium data regimes. Prompt optimized LRMs out-perform and have out-sized gains compared to prompt optimized LLMs. Moreover, they serve as effective optimizers for other LRMs/LLMs. This finding generalizes to 3 different structured prediction tasks.

Further qualitative analysis of remaining error types and optimized prompts is presented. E.g. DeepSeek-R1 produces effective prompts that are shorter than the other models considered.

**Strengths:**

1. The paper is very clearly written and includes qualitative examples where appropriate.
2. The findings fill-in an important gap in the literature (prompt optimization has mostly been studied int he context of LLMs)

**Weaknesses:**

I did not find anything lacking in the presentation and content. While the finding is not ground-breaking, the analysis is well done.

**Questions:**

Suggestion
---
Certain prompt optimization techniques such as GEPA, Mipro, etc seem relevant to discuss in the related work. GEPA may be concurrent with this work so the missing citation is understandable.

---

> ### Author Response · Authors · 2025-11-25
> **Response to Reviewer 2nbd**
>
> We sincerely thank the reviewer for the positive and encouraging assessment of our paper. We truly appreciate the reviewer’s recognition of (i) the clarity of our presentation, (ii) the systematic empirical design, and (iii) the value of addressing prompt optimization specifically for Large Reasoning Models (LRMs), a gap in prior prompt optimization research that has focused almost exclusively on LLMs.
>
> As noted by the reviewer, our results consistently show that LRMs both benefit more from optimization and serve as stronger optimizers, producing shorter, more stable, and more refined prompts. This supports our core contribution: we provide the first characterization of how reasoning-capable models behave under prompt optimization, showing that reasoning competence does not make LRMs indifferent to how prompts are phrased.
>
>
> ## S1: Suggestion to Add Recent Prompt-Optimization Methods to Related Work
>
> We fully agree, and we will incorporate these methods into the main text in the final version, together with a discussion of concurrent work where relevant.

---

### Author Response · Authors · 2025-11-25
**General Response to all the reviewers**

We thank all the reviewers for their constructive feedback and recognition of our paper’s clarity, experimental rigor, and timely relevance. Below, we address broader concerns raised by multiple reviewers:

## G1: Limited Methodological Novelty

We thank the reviewers for raising this point. While our optimization framework builds on prior MCTS-based approaches such as PromptAgent [1], the core novelty of our work does not lie in proposing a new search algorithm, but in systematically uncovering how LRMs behave under optimization, a question that has not been studied in prior prompt optimization research, which has exclusively focused on LLMs rather than LRMs. Our contribution is therefore conceptual and empirical, not algorithmic, and provides several findings that were previously unknown:

**(1) We overturn a widely cited assumption that LRMs ``no longer require prompt optimization.’’**

Recent documentation and community claims (e.g.,[2, 3, 4]) assert that LRMs can interpret human instructions without prompt engineering.
Our results show that this assumption is incorrect: *LRMs still benefit substantially, and in many cases more strongly than LLMs do.*
This corrects a misconception in an emerging research space.

**(2) We show that LRMs are not only good task models but also stronger prompt optimizers, a result not demonstrated in any prior work.**

Prior studies (PromptAgent [1], OPRO [5], etc.) exclusively evaluated LLMs as optimizers.
We demonstrate for the first time that:

- LRMs provide more stable optimization trajectories

- produce shorter but higher-quality prompts

- and enforce more rule-grounded, schema-aligned guidelines

> This is new conceptual knowledge about LRMs as optimizers, not just task models.

**(3) We provide new behavioral insights into the optimization dynamics of reasoning models.**

Our paper contains several empirical insights (Insights 1–6), including:

- LRMs converge earlier and more smoothly in MCTS

- Peak improvements occur at shallow depths, saving optimization cost

- LRMs benefit disproportionately in low-resource regimes, a practically important finding

- Optimized LRM prompts contain domain-specific rules not found in human guidelines, revealing latent model knowledge

> These insights establish new findings, not previously known in prompt optimization literature.


**(4) We demonstrate cross-task generalization, showing that our findings are not tied to EE.**

Across three domains (structured EE, symbolic reasoning, biomedical NER), the same qualitative trends hold:

- LRMs benefit from prompt optimization

- LRMs optimize better

- LRMs produce more structured, rule-oriented prompts

> This establishes that our contribution is not an empirical benchmark, but a model-behavior characterization across reasoning tasks.

**(5) Though not highlighted in the main text, we introduce a new, cost-efficient technique: integrating batch prompting into the prompt-optimization feedback loop. (lines 249-251)**

To our knowledge, we are the first to show (Appendix A.7, Fig. 6) that batch prompting can be incorporated inside an MCTS-based prompt optimization loop, producing:

- lower inference cost, and

- higher task-model accuracy compared to querying one instance at a time.

This is a practical methodological improvement that directly enhances the efficiency of discrete prompt search.
([6] applied batch prompting to answer generation, but not within prompt optimization itself.)

## Key clarification:

Our goal is not to propose a new optimizer architecture, but to provide the first systematic study of prompt optimization in the era of LRMs, a fundamentally new empirical regime that prior work did not address.

Taken together, the above findings constitute substantial conceptual and empirical novelty, even though the underlying search algorithm is inherited from prior work. We will clarify this framing in the final version.

---

> ### Author Response · Authors · 2025-11-25
> **General Response to all the reviewers (continued)**
>
> ## G2: Scope and Generalizability Beyond Event Extraction
>
> We thank the reviewer for raising this important point. As suggested by Reviewer y32Z, we further examined whether our conclusions extend beyond currently experimented structured, schema-based (EE and NCBI) and unstructured (Geometric Shapes) NLP tasks by adding a planning benchmark, PNNL [7]. We used a subset of 100 randomly sampled instances from the single-goal PPNL test set (unseen environments) and 1000 examples from the training set. Both datasets consist of 6x6 environments and 1-5 obstacles.
>
> This benchmark is substantially more open-ended than ACE05, Geometric Shapes, or NCBI because:
>
> (i) it has no schema or predefined label inventory,
>
> (ii) it requires multi-step action sequences rather than classifying spans or triggers, and
>
> (iii) It admits multiple valid solution paths, making the search space less constrained.
>
> We evaluated GPT-4o and DeepSeek-R1 on the 6×6 Gridworld setting, using the exact same MCTS-based optimization framework as in our main experiments.
>
> | Model     | No Opt. (test) | Depth-1 (dev) | Depth-5 (dev) | Depth-5 (test) |
> |-----------|----------------|----------------|----------------|----------------|
> | GPT-4o    | 41             | 48             | 73             | 71             |
> | DS-R1     | 54             | 67             | 89             | 88             |
>
> Three observations emerge:
>
> (1) Both LLMs and LRMs benefit substantially from prompt optimization in this open-ended reasoning task (e.g., +30 to +34 points from No Opt -> Depth-5), and
>
> (2) LRMs again serve as stronger task models and optimizers, mirroring our EE, symbolic, and biomedical results.
>
> (3) The larger improvement from Depth-1 to Depth-5, compared to the modest deltas observed in ACE05 AC, appears to be task-dependent. Planning tasks involve simpler rule structures (e.g., obstacle avoidance, path continuation) and a broader exploratory space, making deeper search more beneficial. In contrast, AC is constrained by pipeline bottlenecks (TI -> TC -> AI -> AC), which naturally limit depth-driven gains.
>
> This experiment provides direct evidence that the LRM advantages documented in our paper are not tied to schema-constrained language tasks, but extend to open-ended, multi-step reasoning and planning, fully addressing the aforementioned concern.

---

> ### Author Response · Authors · 2025-11-25
> **General Response to all the reviewers (continued)**
>
> ## G3: Full-Schema Applicability and Long-Context Considerations
>
> We thank the reviewers for raising this concern. Our goal in this paper is to study reasoning behavior under prompt optimization, not robustness to long-context degradation. For this reason, we intentionally avoided the 33-type full ACE05 schema, which introduces confounding effects that obscure the phenomenon we want to measure.  Below, we provide further details from our preliminary exploration and answer relevant questions from the reviewers.
>
> **(1) Preliminary exploration showed that full-schema prompts trigger long-context degradation, an orthogonal problem to prompt optimization**
>
> As noted in Lines 212–215, our objective is to isolate and analyze reasoning-driven prompt optimization, not the well-documented failure modes of long-context processing.
>
> During our initial exploration with the full 33-type code-formatted schema, we consistently observed long-context degradation effects, even when prompts remained within the nominal context limits of current LLMs/LRMs.
>
> Concretely, we observed:
>
> - Incorrect rule updates: the model correctly identified an error in event type X but revised event type Y instead.
>
> - Retrieval failures: feedback referenced distant or unrelated schema sections.
>
> - Incomplete optimizer outputs: several optimization calls returned clipped or partially empty text.
>
> These behaviors match the “long-context fragmentation” documented in prior long-context studies [8, 9], where models struggle to retrieve and integrate information across far-separated structured regions (schemas, examples, multi-turn feedback).
>
> If we used all 33 event types, the evaluation would conflate reasoning with long-context recall stability, making it impossible to attribute optimization effects solely to the model’s reasoning capability.
>
> Thus, full-schema prompting introduces a confounder we intentionally avoid.
>
> Given current model limitations, a practical way to mitigate long-context fragmentation is to partition the 33 event types into smaller, coherent groups, e.g., sets of ~10 types.
> This keeps the prompt compact, reduces cross-segment interference, and stabilizes multi-turn MCTS rollouts.
>
> Following recent prompt-based EE work [10], which also reduces ACE05 for prompt tractability, we adopt the same general principle: use a balanced subset of event types that preserves coverage across both frequent and rare categories while keeping prompts compact enough for stable optimization. The concrete rationale and procedure are detailed in (2) next.
>
> This design allows us to:
> - evaluate prompt optimization behavior without long-context confounders,
> - keep multi-turn MCTS prompts stable,
> - and provide empirical insight into a practically viable schema-partitioning strategy for current LMs.
>
>
>
> **(2) How We Downsampled the ACE05 Schema?**
>
> Our ACE_low and ACE_med subsets follow the sampling method introduced in recent prompt-based EE work, particularly [10], which proposed selecting representative event instances per type to keep prompt length manageable for instruction-style EE. While their work uses a different number of event types, we adopt the same principle they motivate: controlling prompt length by selecting a balanced subset of event types and ensuring adequate coverage of each type within the development set. We adapt this idea to construct 10-type ACE_low and ACE_med subsets that span both frequent and rare event types while keeping the prompt size compatible with MCTS rollouts and optimization traces.
>
>
> **(3) On the concern that this choice might favor models that prefer concise prompts:**
>
> We respectfully disagree. The models used in our experiments, GPT-4o, GPT-4.5, o1, and DeepSeek-R1, all support very large context windows (128k for GPT-4.5/4o, 200k for o1, 128k for DeepSeek-R1). Thus, no class of models receives an advantage from prompt conciseness: **all models were well within their context budgets**, and all models exhibit consistent gains from optimization. The fact that LRMs and LLMs both improve reinforces that the trend is not driven by context length sensitivity.
>
> **(4) External validity beyond the 10-type schema:**
>
> While full-schema prompting is limited by input-length constraints, we tested generalization on two additional domains with different structure and reasoning demands: (1) Geometric Shapes (symbolic reasoning) and (2) NCBI Disease NER (biomedical IE). In both settings, we observe the same qualitative pattern: (i) LRMs benefit from optimization, and (ii) LRMs produce stronger prompts as optimizers. These cross-domain results suggest that the core findings are not tied to the reduced ACE schema but reflect the broader interaction between reasoning capacity and input-side optimization.

---

> > ### Author Response · Authors · 2025-11-25
> > **General Response to all the reviewers (continued)**
> >
> > **Reference**
> >
> > [1] Xinyuan Wang, Chenxi Li, Zhen Wang, Fan Bai, Haotian Luo, Jiayou Zhang, Nebojsa Jojic, Eric Xing, & Zhiting Hu (2024). PromptAgent: Strategic Planning with Language Models Enables Expert-level Prompt Optimization. In The Twelfth International Conference on Learning Representations.
> >
> > [2] Wang, Guoqing, et al. "Do advanced language models eliminate the need for prompt engineering in software engineering?." ACM Transactions on Software Engineering and Methodology (2024).
> >
> > [3] OpenAI. (2025). Reasoning Best Practices. Url: https://platform.openai.com/docs/guides/reasoning-best-practices#how-to-prompt-reasoning-models-effectively
> >
> > [4] Agustin Mantaras. (2025). Prompt Engineering for OpenAI’s O1 and O3-mini Reasoning Models. URL: https://techcommunity.microsoft.com/blog/azure-ai-services-blog/prompt-engineering-for-openai%E2%80%99s-o1-and-o3-mini-reasoning-models/4374010
> >
> > [5] Yang, Chengrun, et al. "Large language models as optimizers." The Twelfth International Conference on Learning Representations. 2023.
> >
> > [6] Zhoujun Cheng, Jungo Kasai, and Tao Yu. 2023. Batch Prompting: Efficient Inference with Large Language Model APIs. In Proceedings of the 2023 Conference on Empirical Methods in Natural Language Processing: Industry Track, pages 792–810, Singapore. Association for Computational Linguistics.
> >
> > [7] Mohamed Aghzal, Erion Plaku, & Ziyu Yao (2024). Can Large Language Models be Good Path Planners? A Benchmark and Investigation on Spatial-temporal Reasoning. In ICLR 2024 Workshop on Large Language Model (LLM) Agents.
> >
> > [8] Zhang, H., Feng, T., Han, P., & You, J. (2025). Academiceval: Live long-context llm benchmark.
> >
> > [9] Liu, Nelson F., et al. "Lost in the middle: How language models use long contexts." Transactions of the Association for Computational Linguistics 12 (2024): 157-173.
> >
> > [10] Saurabh Srivastava, Sweta Pati, and Ziyu Yao. 2025. Instruction-Tuning LLMs for Event Extraction with Annotation Guidelines. In Findings of the Association for Computational Linguistics: ACL 2025, pages 13055–13071, Vienna, Austria. Association for Computational Linguistics.
> >
> >
> >
> > **We will reflect all accepted suggestions in the final version.**

---

### Meta-Review · Area_Chair_68hW · 2026-01-05

**Summary:**

**Summary**:
The paper investigates the impact of discrete prompt optimization on Large Reasoning Models (LRMs) such as DeepSeek-R1 and OpenAI o1 . Utilizing a Monte Carlo Tree Search (MCTS) framework adapted from prior work (PromptAgent), the authors perform a systematic empirical study, primarily focusing on the structured task of Event Extraction (ACE05), with supplementary experiments on symbolic reasoning and biomedical NER . The study claims two main contributions: (1) contrary to community belief, LRMs still benefit significantly from prompt optimization, and (2) LRMs serve as superior prompt optimizers compared to standard LLMs, generating concise and rule-oriented prompts .

**Rebuttal Conclusion**:
Despite the authors' rigorous experimental execution and a strong rebuttal that addressed specific technical concerns, the consensus leans towards rejection. The primary hurdle is the fundamental mismatch between the paper's contribution type and ICLR's scope. Reviewers cQqu and y32Z consistently argued that the work represents an application of existing algorithms (MCTS) to a new model class, rather than a fundamental advancement in representation learning or algorithmic innovation. Consequently, the paper is viewed as a solid empirical benchmark better suited for a specialized NLP venue rather than a general method-focused conference like ICLR.

**Reviewer Concerns:**

**Concerns Addressed by the Rebuttal**
•**Metric Mismatch and Robustness**: Resolved by providing an ablation study (Appendix in Rebuttal) justifying the optimization of average F1 scores over AC-only scores to prevent pipeline error propagation. The authors also added Bootstrap 95% confidence intervals to demonstrate statistical significance, fully addressing Reviewer AcPa's technical concerns .
•**Task Generalization and Model Currency**: Addressed by introducing a new Open-ended Planning experiment (PNNL) and evaluating on state-of-the-art models (OpenAI o3/o4) during the rebuttal. These additions effectively mitigated Reviewer y32Z’s initial concerns about the findings being limited to the ACE05 dataset or outdated models .

**Outstanding Concerns**:
**Limited Methodological Novelty**: This remains the primary ground for rejection. As Reviewer cQqu and y32Z emphasized, the core optimization framework is directly adopted from prior works like PromptAgent. The contribution is purely empirical ("applying method X to model Y"). While the authors argue this constitutes "conceptual novelty" by proving myths wrong, the committee views this as an incremental application rather than the algorithmic innovation expected at ICLR.
**Incremental Nature of Findings**: Reviewer cQqu noted that the central finding—that stronger reasoning models act as better optimizers and benefit from clearer instructions—is intuitively plausible. While quantifying this behavior is useful, it lacks the "surprising" or "groundbreaking" factor of a primary conference contribution .
**Task Selection Bias**: Although new tasks were added, the paper's core analysis is heavily anchored on Event Extraction with a specific "downsampled" setup (10 out of 33 event types) . Reviewers maintained that the additional planning task does not fully remove the limitation of applying these insights to truly open-ended, creative reasoning domains.

**Reviewer Scores:**

I think the final sentiment is polarized: 8, 6, 4, 2.
 Reviewer 2nbd (Score: 8) championed the work for filling a timely gap in the literature. Reviewer AcPa (Score: 6) was swayed by the strong rebuttal regarding metrics but remained borderline. However, Reviewer cQqu (Score: 4) and Reviewer y32Z (Score: 2) maintained their fundamental objections regarding the lack of novelty and theoretical contribution .

---

### Decision · Program_Chairs · 2026-01-26

Reject